# Direct *in situ* protein tagging in *Chlamydomonas reinhardtii* utilizing TIM, a method for CRISPR/Cas9-based targeted insertional mutagenesis

**Yuqing Hou**, **Xi Cheng**, **George B. Witman** *

Division of Cell Biology and Imaging, Department of Radiology, University of Massachusetts Chan Medical School, Worcester, Massachusetts, United States of America

* george.witman@umassmed.edu

**Data Availability Statement:** All relevant data are within the paper and its Supporting Information files.

## Abstract

*Chlamydomonas reinhardtii* is an important model organism for the study of many cellular processes, and protein tagging is an increasingly indispensable tool for these studies. To circumvent the disadvantages of conventional approaches in creating a tagged cell line, which involve transforming either a wild-type or null-mutant cell line with an exogenous DNA construct that inserts randomly into the genome, we developed a strategy to tag the endogenous gene *in situ*. The strategy utilizes TIM, a CRISPR/Cas9-based method for targeted insertional mutagenesis in *C. reinhardtii*. We have tested the strategy on two genes: *LF5/CDKL5*, lack of which causes a long-flagella phenotype, and *Cre09.g416350/NAP1L1*, which has not been studied previously in *C. reinhardtii*. We successfully tagged the C-terminus of wild-type *LF5* with the hemagglutinin (HA) tag with an efficiency of 7.4%. Sequencing confirmed that these strains are correctly edited. Western blotting confirmed the expression of HA-tagged LF5, and immunofluorescence microscopy showed that LF5-HA is localized normally. These strains have normal length flagella and appear wild type. We successfully tagged the N-terminus of Cre09.g416350 with mNeonGreen-3xFLAG with an efficiency of 9%. Sequencing showed that the tag region in these strains is as expected. Western blotting confirmed the expression of tagged protein of the expected size in these strains, which appeared to have normal cell size, growth rate, and swimming speed. This is the first time that *C. reinhardtii* endogenous genes have been edited *in situ* to express a wild-type tagged protein. This effective, efficient, and convenient TIM-tagging strategy promises to be a useful tool for the study of nuclear genes, including essential genes, in *C. reinhardtii*.

## Introduction

The green alga *Chlamydomonas reinhardtii* provides an excellent model system for the study of many cellular processes [1, 2], and protein tagging is becoming an increasingly indispensable tool for these studies [3–5]. Among other things, protein tags facilitate isolation of the

**Funding:** This work was supported by National Institutes of Health (https://www.nih.gov/) grant R35 GM122574 (GBW), by the Robert W. Booth Endowment at the University of Massachusetts Chan Medical School (GBW), and by grant CDKL5-20-C-101-07 from the LouLou Foundation (https://www.louloufoundation.org/) (YH). The funders had no role in study design, data collection and analysis, decision to publish, or preparation of the manuscript.

**Competing interests:** The authors have declared that no competing interests exist.

tagged proteins and their interacting partners, quantitation of protein amounts by immunoblotting, localization of the proteins by immunofluorescence microscopy, and observation of the dynamic movement and transport of the proteins. The traditional method for creating a *C. reinhardtii* strain expressing a tagged protein involves transformation of either a wild-type strain [6, 7] or a null-mutant strain [8, 9] with a DNA construct that expresses the tagged protein. However, there are drawbacks to this approach. First, when a wild-type strain is used, the expression of endogenous wild-type protein may compete with the tagged protein and cause ectopic localization of the latter, potentially giving a misleading result [6]. Second, when the protein of interest is essential for viability, a null mutant will not be available for tagging experiments; indeed, a null mutant may not be available even for a non-essential gene. Third, the approach is time consuming and laborious if a null mutant must first be generated. Here, we detail a new approach, based on our previously described Targeted Insertional Mutagenesis (TIM) method [10], that circumvents these difficulties by allowing direct *in situ* tagging of specific genes.

For over two decades, insertional mutagenesis has been an important method for generating *C. reinhardtii* mutants [11–13]. In conventional insertional mutagenesis (Fig 1A), *C. reinhardtii* cells are transformed with a fragment of DNA containing a selectable marker, which then inserts randomly into the genome, creating a mutation wherever it inserts. These mutations generally involve large genomic changes, and thus usually are null mutations, or truncation mutations in which part of the gene upstream or downstream of the insertion is still expressed. Because genes are mutated randomly, luck and considerable effort are required to identify a mutant in which a specific gene of interest has been disrupted. This shortcoming has been addressed in part by the creation of a large genome-wide indexed library of *C. reinhardtii* insertional mutants [13]. While this library has proven to be very useful to many investigators, not all genes are represented in the current library, the parent strain has a phenotype that may compromise the mutant strains' utility for some types of experiments, and strains must be crossed if it is desirable to have the mutation in a specific genetic background.

More recently, we described TIM, a universal, highly efficient, and relatively simple CRISPR/Cas9-based gene-editing protocol for targeted disruption of *C. reinhardtii* genes [10]. TIM is based on insertional mutagenesis. However, in contrast to conventional insertional mutagenesis, the TIM method allows a specific gene to be targeted for disruption (Fig 1B). Cells are transformed with a Cas9-guide RNA (gRNA) ribonucleoprotein (RNP) and donor DNA containing a selectable marker and short gene-specific homology arms at its ends. Once in the cell, the RNP generates a double-strand break (DSB) at the target site, which, together with the homology arms, facilitates insertion of the donor DNA at that site, so that insertion there becomes much more probable than insertion at a random site. In our hands, the percent of transformants with insertions at the target site typically have ranged from 40% to 90% [10]. Following selection of transformants, strains containing insertions at the target site can be easily identified by PCR analysis. As with conventional insertional mutagenesis, the colonies identified in this way are usually null mutants. TIM has considerable utility, as it is likely to be applicable to any non-essential *C. reinhardtii* gene, it allows an investigator to begin with a strain of choice (including cell-walled strains), it can utilize different selectable markers, and it can be used to generate double mutants in a single step.

In the present study, we develop a related strategy, based on TIM, that enables direct, highly efficient tagging of specific *C. reinhardtii* genes *in situ*. We term the new method "TIM-tagging" (Fig 1C). The strategy utilizes CRISPR/Cas9 gRNA to generate a DSB at an appropriate spot in the target gene, and a donor DNA encoding 1) part of the target gene with a tag, 2) a selectable marker, and 3) gene-specific homology arms to facilitate precise insertion of the donor DNA into the target site. The gRNA cut site is chosen so that it is downstream of the

**Fig 1. Schematic overview of conventional insertional mutagenesis, TIM, and TIM-tagging in *C. reinhardtii*.** (A) Conventional insertional mutagenesis to create mutants randomly. Donor DNA encoding a selectable marker is introduced into the cell; it then inserts randomly into the nuclear genome, causing a large insertion, deletion, and/or reorganization at the site of integration. (B) Targeted insertional mutagenesis (TIM) to disrupt a specific gene of interest. Based on insertional mutagenesis, but a CRISPR/Cas9 RNP is included in the transformation step and short gene-specific homology arms are added on both ends of the donor DNA. The RNP creates a DSB at the target site, which facilitates the incorporation of donor DNA at this site. The homology arms further increase the probability that the donor DNA will insert at this site, but they are not critical for success. Transformants expressing the selectable marker are screened by gene-specific PCR analysis to identify those in which the target gene has been disrupted. (C) TIM-tagging to specifically tag a gene of interest, as illustrated here for C-terminal tagging. Based on TIM, but the RNP target site is placed close to and upstream of the intended tagging site. The donor DNA starts with an essential homology arm to promote precise integration via homology-directed repair (HDR) at this end of the donor DNA, and then continues with the rest of the gene, including the tag sequence, followed by a selectable marker. A second homology arm at the 3' end of the donor DNA is optional but might be helpful. In the final result, the allele contains the tag sequence, and is followed by the selectable marker. Following selection of transformants, cells that have integrated the tag sequence are identified by gene-specific PCR analysis. Green star: start codon; red star: stop codon; black and white boxes: homology arms on donor DNA or sequence on gene matching homology arms.

site where the tag sequence is intended to be for N-terminal tagging, or upstream of the site where the tag sequence is intended to be for C-terminal tagging. The donor DNA is designed to place the selectable marker upstream of the 5' end of the gene or downstream of the 3' end of the gene for N- and C-terminal tagging, respectively. We demonstrate the feasibility of this new strategy by successfully tagging LF5/CDKL5, a protein involved in flagellar length regulation [14], at its C-terminus with a 3HA tag [15], and Cre09.g416350, which is homologous to human nucleosome assembly protein 1 like 1 (NAP1L1), at its N-terminus with a

mNeonGreen-3xFLAG tag [16]. Of all selected transformants in these experiments, 7–9% were successfully tagged.

This is the first time that *C. reinhardtii* genes have been tagged by editing the endogenous alleles directly *in situ*. This TIM-tagging strategy has the following benefits: 1) the edited gene is expressed under its own endogenous promoter and with its own 3' UTR; 2) there is no need to acquire a null mutant as a prerequisite; 3) there is no wild-type protein to interfere with the tagged protein; 4) the strategy is flexible in that various tags, drug-resistance cassettes on donor DNA, and *C. reinhardtii* starting strains can be chosen to suit a researcher's specific needs. Thus, it promises to be a very useful tool for tagging nuclear genes, including essential genes, in *C. reinhardtii*.

A preliminary report on this research was presented at Cell Bio Virtual 2021, a joint online meeting of the American Society for Cell Biology and the European Molecular Biology Organization [17].

## Materials and methods

### Strains and media

*C. reinhardtii* strains utilized in this study were g1 (CC-5415; *nit1*, *agg1*, *mt+*), CC-4560 (*lf5-2*, *mt+*), CC-620 (*nit1*, *nit2*, *mt+*), and CC-621 (*nit1*, *nit2*, *mt-*) (Chlamydomonas Resource Center, https://www.chlamycollection.org/). g1 was used for TIM-tagging to create HA-tagged LF5 strains and mNeonGreen-3xFLAG-tagged NAP1L1 strains. g1 was also used as a wild-type control as needed. CC-4560 was used in Figs 5A and 6B as an *lf5* null mutant control. CC-620 and CC-621 were used to produce autolysin required for the transformation protocol. Media used were: Tris-acetate-phosphate (TAP) medium [18]; M (minimal) medium I [19] modified to contain 0.0022 M $KH_2PO_4$ and 0.00171 M $K_2HPO_4$ [20]; M-N medium (modified M medium lacking $NH_4NO_3$).

### Guide RNA and donor DNA

Guide RNAs were designed using the online custom Alt-R® CRISPR-Cas9 guide RNA tool of Integrated DNA Technology (IDT) (https://www.idtdna.com/site/order/designtool/index/CRISPR_CUSTOM). The number of non-specific binding sites for each gRNA was checked using the CRISPR-direct website (http://crispr.dbcls.jp) [21] and served as a second criterion in selecting the gRNA.

For LF5 tagging, donor DNA was amplified by PCR from a plasmid template (pLF5HA) using primer pair LF5donor-1/LF5donor-2. To construct pLF5HA, plasmid pLF5CsfGFP was first constructed by piecing together four fragments using NEBuilder® HiFi DNA Assembly Master Mix (catalog number E2621) (S1A Fig). The four fragments were: 1) PCR product encoding the *LF5* gene up to but not including its stop codon, amplified from pBS3830 [14] using primer pair LF5-1/LF5-6; 2) PCR product encoding the remainder of the *LF5* gene, amplified from pBS3830 using primer pair LF5-4/LF5-5; 3) the sfGFP cassette amplified from plasmid pIFT140-sfGFP-aphVIII [22] using primer pair sfGFP-1/sfGFP-2; and 4) plasmid pKS-aphVIII-lox [23] linearized by SacI. Plasmid pLF5CsfGFP except for its sfGFP region was then amplified using primer pair LF5-20/LF5-31 to yield fragment pLF5CsfGFPΔsfGFP, which was pieced together, again using NEBuilder® HiFi DNA Assembly Master Mix, with the 3HA fragment cut from p3HA [15] by StuI and NheI to yield pLF5HA (S1B Fig). The resulting plasmid pLF5HA thus contains a full-length *LF5* gene with sequence encoding the 3HA tag immediately before the stop codon, and the *aphVIII* gene at the 3' end of the *LF5* sequence.

For NAP1L1 tagging, donor DNA was amplified from plasmid pNAP1L1NmNeon3Flag using primer pair NAP1L1donor-1-2/NAP1L1donor-2-2. To generate plasmid

pNAP1L1NmNeon3Flag, NAP1L1 gene fragments were amplified from purified g1 genomic DNA using the primer pairs NAP1L1-3/NAP1L1-4 and NAP1L1-5/NAP1L1-6. The mNeon-Green-3xFLAG tag was amplified from plasmid pLM160 [16] using primer pair PLM160-1/PLM160-2. These fragments were joined together with KpnI-cut pHyg3 [24], again using NEBuilder® HiFi DNA Assembly Master Mix, to yield the plasmid pNAP1L1NmNeon3Flag.

See S5 Appendix for pLF5CsfGFP, pLF5HA, and pNAP1L1NmNeon3Flag plasmid sequences. All primers used in this study are listed in S1 Table.

## Preparation of Cas9/gRNA RNP

*LF5* gRNA was from IDT [Alt-R® CRISPR-Cas9 sgRNA (2 nmol)]. *NAP1L1* gRNAs were from Invitrogen [TrueGuide™ Synthetic sgRNAs (1.5 nmol)]. Guide RNA was dissolved in Nuclease-Free Duplex Buffer (IDT) to a final concentration of 20 µM. To make Cas9/gRNA RNP, two microliters of 20 µM sgRNA were incubated with 5 µg of IDT Alt-R® S.p.Cas9 Nuclease V3 in IDT RNA duplex buffer at 37°C for 15 minutes in a final volume of 10 µL. All CRISPR gRNAs used in this study are listed in S1 Table.

## Cas9/gRNA *in vitro* assay

NAP1L1 DNA substrate was amplified from purified wild-type genomic DNA by PCR with primer pair NAP1L1-25/NAP1L1-22 using Herculase II Fusion DNA Polymerase (Agilent, catalog number: 600675). RNP was made as described above. *In vitro* digestion reactions were run in a 20-µl volume with 150 ng DNA substrate and respective RNP at a molar ratio of 1:10 in 1×NEB buffer 3.1 for 30 minutes at 37°C. The samples were then treated for 5 minutes at 100°C, allowed to cool slowly, and fragments separated on a 1.2% tris-acetate-EDTA agarose gel.

## Delivery of Cas9/gRNA RNP and donor DNA

Preparation of autolysin and delivery of Cas9/gRNA RNP and donor DNA into g1 cells was done as described [10]. Briefly, cells grown on TAP plates were harvested and treated with autolysin to remove cell walls. Following cell wall removal, cells were incubated at 40°C for 30 minutes with gentle agitation. Cells were then washed and resuspended in TAP medium + 2% sucrose at a concentration of 2.0–7.0x10^8 cells/mL. Ten µL of Cas9/gRNA RNP (or 10 µL of each Cas9/gRNA RNP in the case of the two-RNP experiment) and 1.5 µg of donor DNA were mixed with approximately 110 µL of concentrated cells to give a final volume of 125 µL. The RNP and donor DNA were then delivered into the cells by electroporation. The cuvette was then incubated immediately at 16°C for 1 h, after which the cells were transferred to 10 mL fresh TAP medium + 2% sucrose and gently rocked under dim light for 24 h at room temperature. The cells were then collected and plated, using top agar as described [10], onto TAP medium + 1.5% agar plates containing 10 µg/mL paromomycin or hygromycin. Plates were maintained on a 14:10 light:dark cycle at 23°C until colonies were big enough to pick.

## Genotyping and sequencing of potentially tagged strains

Genotyping by PCR and sequencing of potentially tagged strains were done as described in [10]. For LF5 tagging, primer pair LF5-37/LF5-28 was used for the initial screening. Primer pair LF5-37/LF5-30 was used to check the integrity of the 3' UTR of the *LF5* gene. Primer pair Fus-3/Fus-4, amplifying a portion of the mating-type-plus-specific gene *FUS1* [25], was used as a positive control. Primer pair LF5-28/AphVIII-3 was used to amplify sequence surrounding the site where the 3' end of the donor DNA integrated. The purified PCR products were

sent for sequencing using the primers that amplified them. For longer PCR products amplified during initial screening, primers AphVIII-2, AphVIII-1, and LF5-39 also were used to sequence the ends as required. For PCR products amplified with primer pair LF5-28/AphVIII-3, primers LF5-39, AphVIII-4, and AphVIII-1 also were used as required for sequencing. The PCR product amplified from LF5HA92 could not be sequenced directly and was cloned into pGEM®-T easy vector (Promega, Catalog number A1360) and sequenced.

For NAP1L1 tagging, primer pair PLM160-5/NAP1L1-12 was used to screen for correct insertion of the 3' end of the donor DNA into the target site. PCR products were sequenced using primer PLM160-5. Primer pair NAP1L1-19/NAP1L1-18 was used to screen for correct insertion of the 5' end of the donor DNA into the target site. PCR products were sequenced using primer NAP1L1-18.

## Western blotting

Western blotting was carried out as described [10]. Membranes were probed with the following antibodies at the indicated dilutions: rat monoclonal anti-HA (1:2000, clone 3F10, Sigma, catalog number 11867423001), rabbit polyclonal anti-LF5 (1:5000) [14], mouse monoclonal anti-FLAG (1:5000, Sigma, catalog number F1804), and rabbit polyclonal anti-ATP synthase β subunit (1:100,000; Agrisera, Vännäs, Sweden, catalog number AS05 085).

## Immunofluorescence microscopy

Cells were fixed and stained for immunofluorescence microscopy by the alternate protocol of [26] with minor modifications. Specifically, cells grown in 24-well cell culture plates containing liquid M medium were applied to multi-well slides, which had been pre-treated with 1% polyethylenimine for 30 seconds and then rinsed with distilled water, for 3 minutes. The cells on the slide were then fixed with cold methanol at -20°C for 10 minutes. After air drying, the slides were incubated in 1×PBS for 1 h, followed by blocking buffer [5%(w/v) BSA, 1%(v/v) fish skin gelatin, 10%(v/v) goat serum, in 1×PBS] for at least 1 h. Slides were then incubated in blocking buffer containing primary antibodies [rabbit monoclonal antibody to HA-Tag (C29F4, Cell Signaling catalog number 3724) diluted 1:200, and mouse monoclonal anti-acetylated tubulin antibody (Sigma-Aldrich, Inc., clone 6-11B-1, catalog number T-6793) diluted 1:2000] overnight at 4°C. The slides were then washed four times with blocking buffer over a period of 1 h. The slides were next incubated in blocking buffer containing secondary antibodies [F(ab')2- Goat anti-Rabbit IgG (H+L) Cross-Adsorbed Secondary Antibody, Alexa Fluor™ 488, Thermo Fisher A11070; and F(ab')2-Goat anti-Mouse IgG (H+L) Cross-Adsorbed Secondary Antibody, Alexa Fluor™ 568, Thermo Fisher A11019] diluted 1:2000 for 1 h in darkness. Slides were then washed 2 times with blocking buffer and transferred to a large volume of 1×PBS for a final wash. The slides were air dried, and coverslips were mounted onto them using ProLong™ Gold Antifade Mountant (Thermo Fisher P36930). Images were acquired at room temperature using AxioVision software and a camera (AxioCam MRm) on a microscope (Axioskop 2 Plus) equipped with a 100×/1.4 oil differential interference contrast (DIC) Plan-Apochromat objective (Carl Zeiss Microimaging, Inc.) and epifluorescence.

## Flagellar length analysis

Cells were synchronized by growing in M medium aerated with 5% $CO_2$ and 95% air under 14 h light/10 h dark cycles. Cells in mid-log phase were fixed during the light phase of the cycle by mixing equal volumes of the cell culture and 2% glutaraldehyde. Cells were imaged at room temperature using AxioVision software and a AxioCam MRm camera on an Axioskop 2 Plus microscope equipped with a 40X objective (Carl Zeiss Microimaging, Inc.) and phase contrast.

Flagellar lengths were measured using image J software (https://imagej.nih.gov/ij/). For each strain, one flagellum from each of 50 randomly picked cells was used for statistical analysis and plotted using GraphPad Prism 9.3.1.

## Results

To tag a protein at the C terminus or the N terminus, we modified the TIM procedure by moving the RNP target cut site to the 3'- or 5'-terminus of the gene, respectively, and by changing the design of the donor DNA.

### Strategy for tagging the C-terminus

To tag a protein at the C-terminus (Fig 2), an RNP-cut site just upstream of the stop codon is selected. Wild-type cells are transformed with both RNP and donor DNA, designed as in Fig 2, using the TIM protocol (https://www.protocols.io/view/tim-a-targeted-insertional-mutagenesis-method-util-bdcki2uw). Antibiotic-resistant cells are then selected by growth on agar, and selected colonies are screened by PCR to identify those likely to have the desired gene edit. Among the positive clones identified in this way, the integrity and the expression of the tagged gene can be confirmed by sequencing the PCR product and by western blotting.

### Strategies for tagging the N-terminus

We envision two closely related strategies for tagging the N-terminus. In the first, which utilizes a single RNP, a target RNP-cut site just downstream of the start codon is selected (Fig 3). The donor DNA is designed as explained in Fig 3 legend. Wild-type cells are transformed with both the RNP and the donor DNA using the TIM protocol (https://www.protocols.io/view/

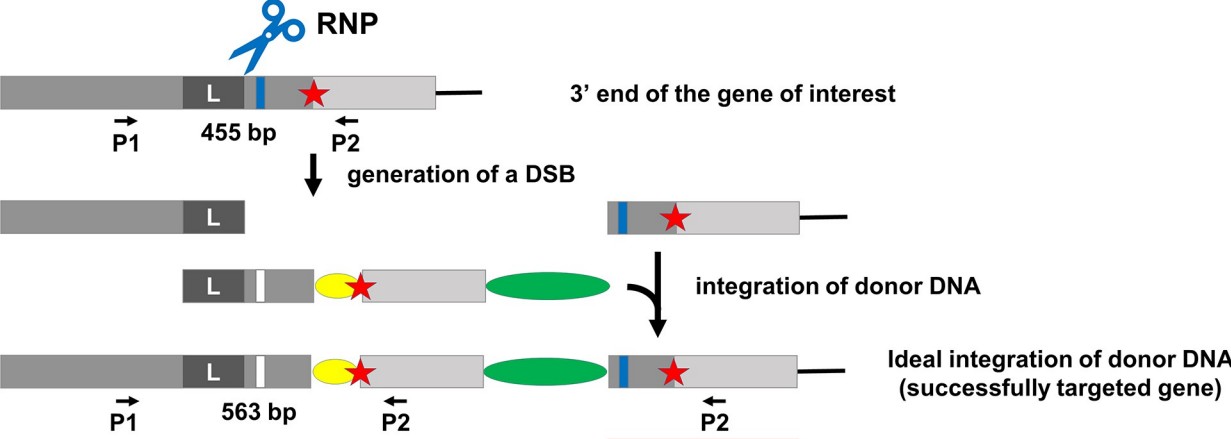

**Fig 2. Schematic overview of TIM-based C-terminal tagging strategy.** The target gene is illustrated at top, with the protospacer adjacent motif (PAM) shown as a blue bar and the stop codon as a red asterisk; the target RNP-cut site is located upstream of the stop codon. Donor DNA is designed to include, successively, a "left" homology arm (L) homologous to sequence upstream of the target cut site, gene sequence downstream of the target cut site, including a PAM with silent mutation (white bar on donor DNA), the desired tag sequence (yellow oval), the stop codon (red star), 3' end non-coding sequence (light grey box), and the drug-resistance gene (green oval). The RNP can cut the wild-type gene, creating a DSB, but not the donor DNA or the tagged gene due to the silent mutation at the PAM site. When, with the help of the left homology arm, the donor DNA is inserted into the cut site without any modification, a tagged wild-type gene is formed as illustrated here. This gene carries the silent mutation at the PAM site. The drug-resistance cassette from the donor DNA is located downstream of the gene. A portion (indicated by a red line) of the original gene remains immediately downstream of the drug-resistance cassette. Following selection of colonies by growth on antibiotic-containing agar, clones are screened by PCR using a 5'-end primer (P1) complementary to sequence upstream of the donor DNA region and a 3'-end primer (P2) complementary to sequence on, or downstream of, the tag. In the latter case (shown here), product amplified from the wild-type gene will be smaller than that from the tagged gene. The sizes of the expected PCR products for the specific experiment shown in Fig 5 are indicated between the primers.

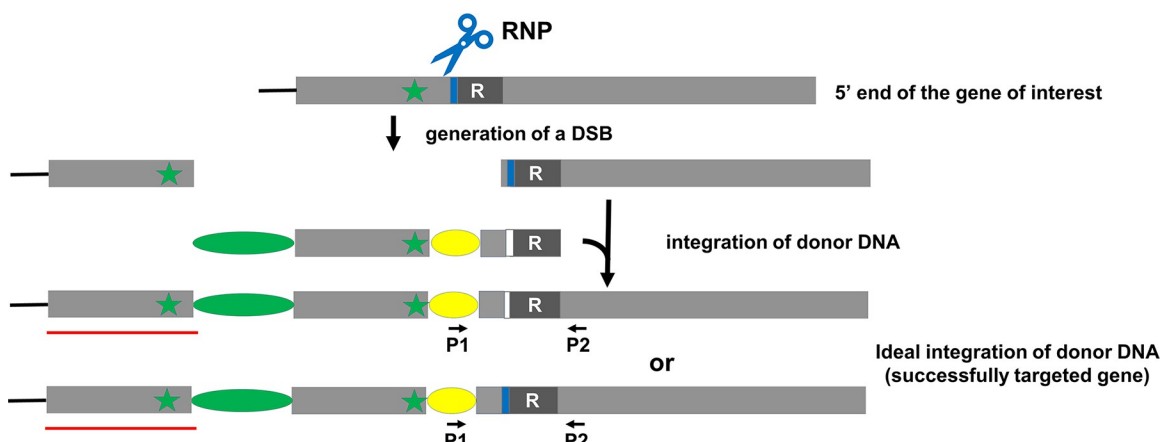

**Fig 3. Schematic overview of TIM-based N-terminal tagging strategy using a single RNP.** The 5' end of the gene is shown as a grey box. The start codon is shown as a green star. The RNP target cut site is located downstream of the start codon. The PAM is shown as a blue bar on the gene. The donor DNA begins with a drug-resistance gene (green oval), then sequence homologous to the 5' end of the target gene and continuing through the start codon, then sequence encoding the desired tag (yellow oval), and finally sequence homologous to that of the target gene beginning immediately after the start codon, including the RNP PAM modified with a silent mutation (white box), and continuing downstream for at least 20 bp (the "right" homology arm). The RNP can cut the wild-type gene, creating a DSB, but not the donor DNA or tagged gene if the donor DNA end is used as template to repair the DSB (upper diagram of "successfully targeted gene"). The new 5' end created by the DSB also can be used as a template for repair, resulting in a tagged allele with a wild-type PAM sequence (lower diagram of "successfully targeted gene"). In these tagged alleles, the drug-resistance cassette is upstream of the gene. A portion (indicated by a red line) of the original gene remains immediately upstream of the drug-resistance cassette. To screen for positive clones, a 5'-end primer (P1) is designed to be complementary to the tag sequence (as illustrated here) or to sequence upstream of the tag; the 3'-end primer (P2) is designed to be complementary to sequence downstream of the homology arm.

tim-a-targeted-insertional-mutagenesis-method-util-bdcki2uw). Following selection of colonies on antibiotic-containing agar plates, selected colonies are screened by PCR to identify those likely to have the desired insertion. As in the case of C-terminal tagging, positive clones should be confirmed by sequencing the PCR products and western blotting.

The second strategy utilizes two RNPs (Fig 4). RNP1 targets a site just upstream of the gene and RNP2 targets a site just downstream of the start codon. Donor DNA is designed as in Fig 4. If inserted as expected, the donor DNA will replace the sequence between the two cut sites. The advantage of this approach is that 1) there will be no duplication of the gene's 5'-regulatory sequence, which can occur when using the above single-RNP strategy for N-terminal tagging, and 2) insertion of the 5' end of the donor DNA, if it occurs by homology-directed repair (HDR), is less apt to result in deletions or insertions that could affect expression of the target gene. Selection and screening of colonies is as described for the single-RNP strategy except that PCR screening also uses a second pair of primers designed to amplify sequence corresponding to the regions surrounding the RNP1 cut site.

## Proof of concept: TIM-tagging LF5 with HA at the C terminus

We tested the 3'-end tagging strategy on the *LF5* gene using the 3HA tag [15] and a paromomycin-resistance cassette. We picked the *LF5* gene because it is involved in flagella length control, and lack of it causes a long-flagella phenotype [14]. Moreover, LF5 has a distinct localization in flagella, concentrating at the proximal end of the flagellar shaft. The mutant phenotype and the distinct localization within the flagellum provide excellent benchmarks for evaluating if the HA-tagged strains express HA-tagged LF5 with wild-type function and localization. While an antibody to LF5 is available [14] and gives satisfactory results when used for western blotting and immunofluorescence microscopy of flagella, it is less satisfactory for

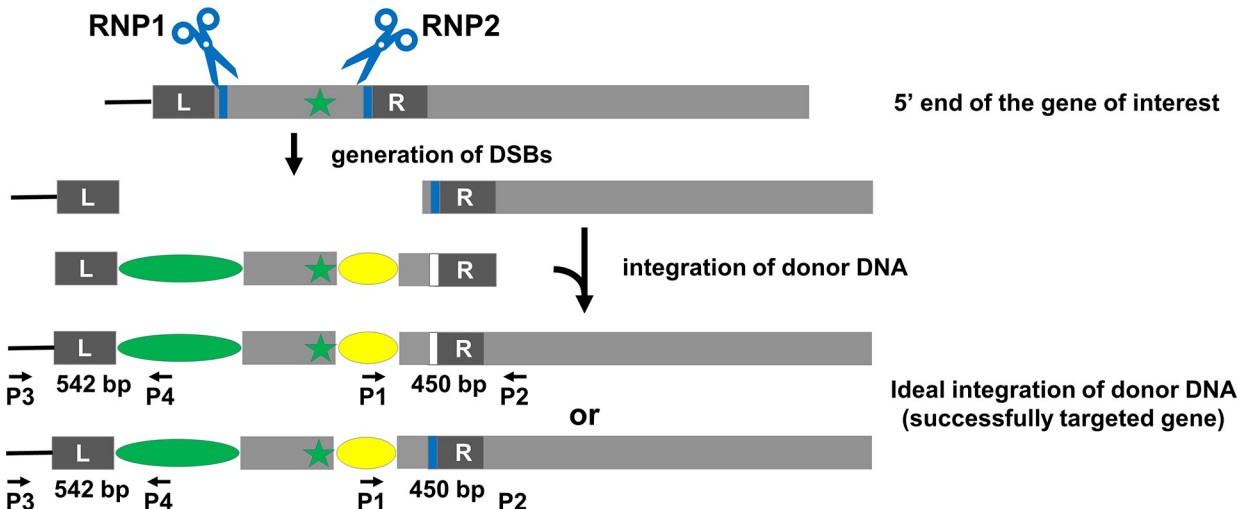

**Fig 4. Schematic overview of TIM-based N-terminal tagging strategy using two gRNAs.** The 5' end of the gene is shown as a grey box, the start codon is indicated by the green star, and the PAMs are indicated by blue bars. RNP1 cuts upstream of, but as near as possible to, the start of the gene, while RNP2 cuts downstream of the start codon. The donor DNA is identical to that of Fig 3 except that it starts with a left arm (L) homologous to genomic sequence just upstream of the RNP1-cut site. In the ideal situation, the DSB created by RNP1 and RNP2 will be repaired by homologous integration of the donor DNA facilitated by the left and right homology arms, resulting in a wild-type gene tagged at its N-terminus, and a drug-resistance gene just upstream of the tagged target gene. During integration, the 3' end of the donor DNA or the genomic DNA's new 5' end created by the DSB can be used as a template, creating either a tagged allele with a mutated RNP2 PAM sequence (top diagram of "successfully targeted gene") or a tagged allele with a wild-type RNP2 PAM sequence (bottom diagram of "successfully targeted gene"), respectively. Following selection of colonies on antibiotic-containing medium, colonies are screened by PCR using primer pairs P1/P2 and P3/P4 to identify those in which the insertion has occurred as planned. The sizes of the expected PCR products for specific experiments of Figs 8 and 9 are shown between the primers.

whole cells. Thus, an HA-tagged strain would facilitate study of LF5 in the cell body by taking advantage of a commercially available anti-HA antibody, which is very specific when used with *C. reinhardtii* cells.

We randomly picked 94 antibiotic-resistant transformants for PCR screening (Fig 5) using primers as illustrated in Fig 2. The wild-type and HA-tagged *LF5* genes should yield PCR products of 455 bp and 563 bp, respectively. Among the 94 clones, 17 yielded PCR product of the size expected to originate from the wild-type gene, and 11 yielded products of the size expected for the HA-tagged gene. All the other clones either yielded no PCR product or product of a size different from the expected sizes, suggesting that in these clones the *LF5* gene was edited, but not as desired. To confirm that the lack of a PCR product from these strains was the result of gene editing rather than errors in the PCR assay, we analyzed these colonies again by PCR using primers to *LF5* and to *FUS1* as a positive control (S2 Fig). The result confirmed that failure to amplify *LF5* in these strains was due to editing events. Therefore, among all drug-resistant transformants, the overall targeting efficiency by CRISPR/Cas9 is high (about 82%), with only 17 out of 94 clones retaining an unedited wild-type gene. This high efficiency is consistent with our previous results for the TIM method [10].

The 11 PCR products of the expected size for correct insertion, as well as six longer PCR products (L1-L6 in Fig 5), were purified and sequenced (S1 Appendix). Ten of the 11 positive clones had the desired edits, with HA inserted and PAM mutated. The eleventh (red "S" in S1 Appendix) was cut at the desired site, but only 108 bp from the 3'-end of the donor DNA was inserted there, resulting in a PCR product of the same size as those of the correctly edited clones. All 6 clones that yielded longer PCR products had drug-resistance cassettes, with modifications at their ends, inserted at the cutting site.

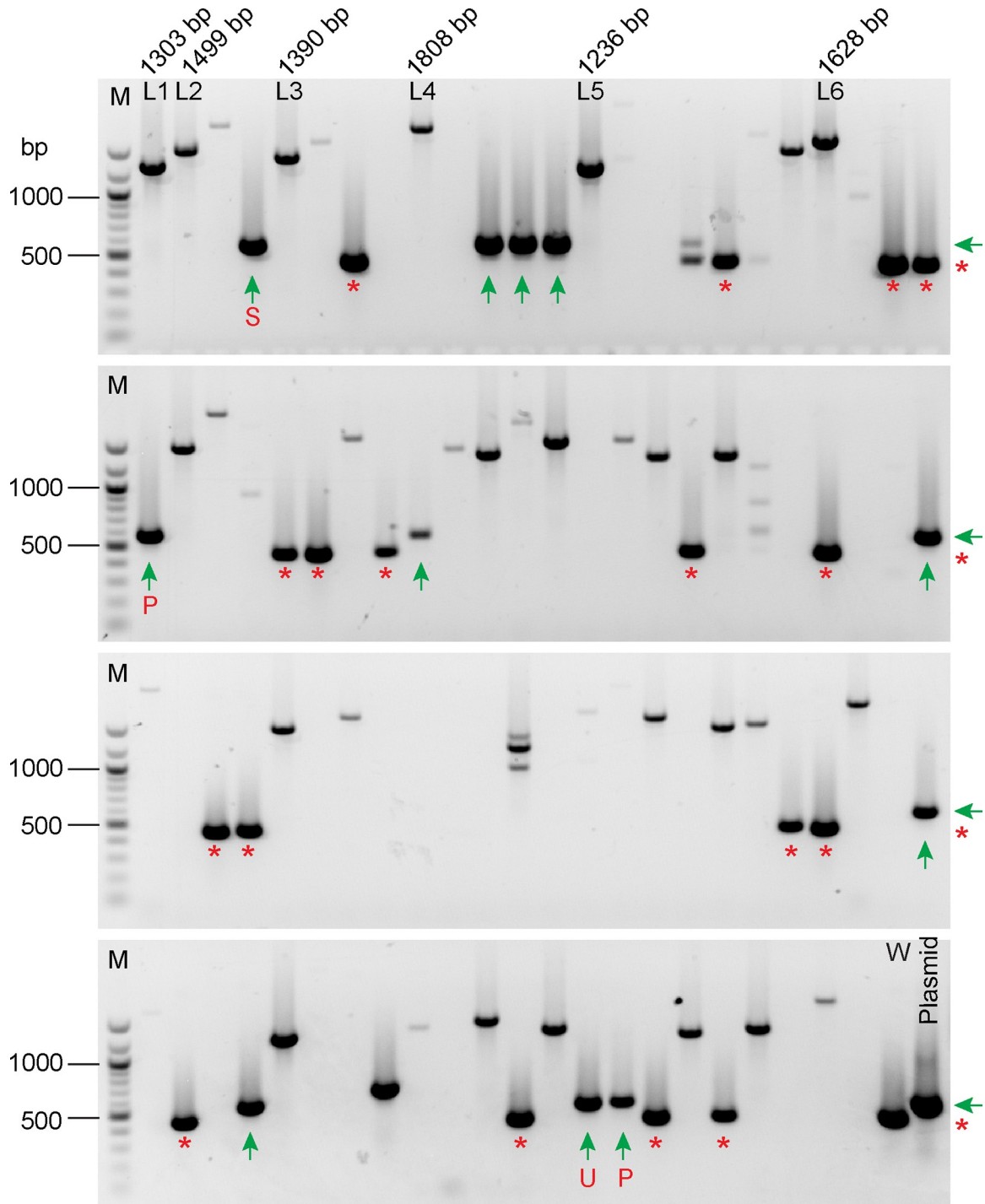

**Fig 5. Initial PCR screening of transformants from the LF5-HA C-terminal tagging experiment.** DNA from the wild-type strain (W), 94 transformants, and the plasmid pLF5HA, which contains the *LF5-HA* gene, were used as templates for PCR screening with primer pairs P1 and P2 as illustrated in Fig 2. The expected size for the wild-type product was 455 bp (marked with red * on the right side of the gels and below the bands). In theory, one small fragment (563 bp) and one large fragment (3576 bp) would be predicted as products resulting from PCR amplification of the tagged gene in this experiment, yet only the small fragment was observed (marked with green arrows on the right side of the gels and below the bands). There are several likely reasons for this. First, we used a PCR elongation time optimal for 563-bp products, so there would not have been enough time for the polymerase to amplify products as large as 3576 bp. Second, to facilitate screening of a large number of clones, we used crude DNA as template, and in our experience it is more difficult to amplify longer PCR products from crude DNA. Third, due to the high GC content of *C. reinhardtii* DNA, it is harder to

amplify long PCR products than short ones. The presence of the wild-type product indicates that the cell line was not edited at the target site, and the antibiotic-resistance gene has inserted into the genome somewhere else. The absence of the wild-type product indicates that the cell line was edited at the RNP-cut site. Among these 94 cell lines, 11 yielded products of the size expected from the tagged gene, suggesting that the donor DNA was inserted as desired. The PCR products from these 11 strains were sequenced for further confirmation: 10 had the sequence expected for HDR; one (here marked with a red S underneath the band) did not (S1 Appendix). Among the 10 clones with the expected sequence, eight expressed LF5-HA (see Fig 6); only two (here marked with a red P) did not. Among the eight clones that expressed LF5-HA, all but one (here marked with a red U) had a complete 3' UTR (see Fig 7). The complete absence of PCR product or product of a size other than that expected from the wild-type or tagged gene indicates that a more complicated editing event occurred. These are likely to be *lf5* mutants. Six products (L1-L6), the sizes of which range from 1235 bp to 1808 bp, are different from the large fragment (3576 bp) expected from an ideal integration of donor DNA. L1 to L6 were the products of alleles in which only a part of the donor DNA, consisting mainly of the drug-resistance cassette, had integrated into the gene (S1 Appendix). Nucleic acid markers (M) (NEB 100 bp DNA ladder, catalog number NO551), from top to bottom, are 1517, 1200, 1000, 900, 800, 700, 600, 500/517, 400, 300, 200, and 100 bp.

We then tested the 10 correctly edited clones by western blotting to see if LF5-HA protein was expressed. Eight out of 10 expressed normal levels of LF5-HA as assessed using anti-HA antibody and anti-LF5 antibody (Fig 6A). To determine if the LF5-HA protein localized normally, cells were probed with anti-HA and anti-acetylated tubulin antibodies and observed by immunofluorescence microscopy (Fig 6B). LF5-HA protein had the expected localization at the base of the flagellar shaft in all 8 clones.

Because the initial PCR screen did not include all of the *LF5* gene's 3' UTR, we examined the integrity of the 3' end of the *LF5* gene in the above 8 clones using PCR primers designed to amplify a longer product that includes the complete 3' UTR. Seven clones yielded a PCR product of the expected size and intensity (Fig 7A), while one failed, indicating that in this strain the 3' UTR likely is incomplete. The PCR products from the seven positive strains were sequenced and all contained a complete 3' UTR (S1 Appendix). We then checked the phenotypes of these seven clones. Null mutants of *lf5* have extra-long flagella and thus swim abnormally. All seven HA-tagged clones had normal length flagella (Fig 7B) and appeared to swim normally, consistent with the HA-tagged LF5 protein being fully functional.

We next amplified the region surrounding the integration site of the 3' end of the antibiotic-resistance cassette from the eight strains that expressed HA-tagged LF5 protein. PCR and sequencing results (S2 Appendix) showed that three of them have in-del mutations at the integration site, which is typical for non-homologous end joining. The other strains have large insertions, deletions, and reorganizations, which is common for insertional mutagenesis in *C. reinhardtii*.

In summary, from 94-antibiotic-resistant clones screened, we obtained seven that had a complete *LF5* 3' UTR and expressed wild-type LF5 with a C-terminal HA tag, for a tagging efficiency of 7.4%.

## TIM-tagging NAP1L1 with mNeonGreen-3xFLAG at the N terminus

To test if TIM-tagging also can be used to tag a gene at its 5' end, we targeted *Cre09.g416350* using the two-RNP strategy of Fig 4. Because *Cre09.g416350* is homologous to human *NAP1L1*, we will hereafter refer to it as *NAP1L1*. For this test, we chose to use the mNeonGreen-3xFLAG tag [16] and the hygromycin-resistance gene as the selectable marker. Following transformation of wild-type cells with the two RNPs and the donor DNA, we randomly picked 96 hygromycin-resistant transformants and analyzed them by PCR. PCR using primer pair P1/ P2, which amplifies sequence encompassing the RNP2 target site, as illustrated in Fig 4, yielded a product of the expected size from 16 of the colonies (Fig 8). Sequencing of these 16 PCR products (S3 Appendix) showed that the DSB was correctly repaired with donor DNA inserted in 12 of them as expected for HDR. Among the 12 clones, eight have the PAM mutation introduced by the donor DNA and four do not, indicating that during the repair process,

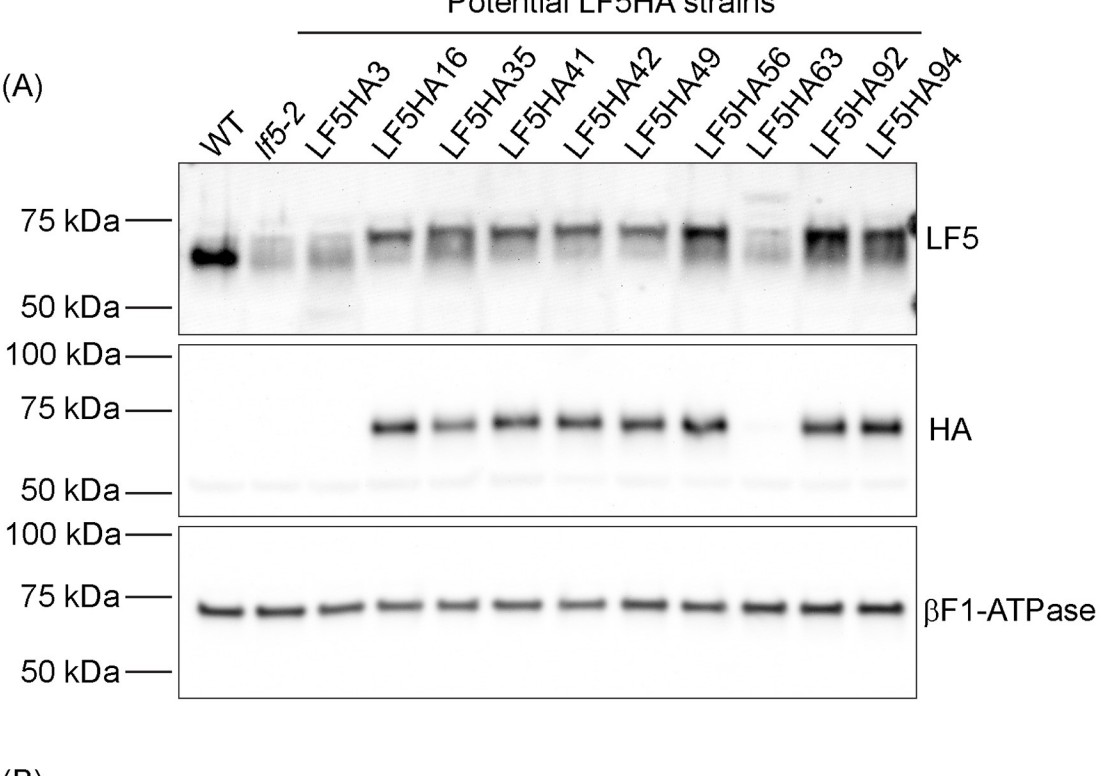

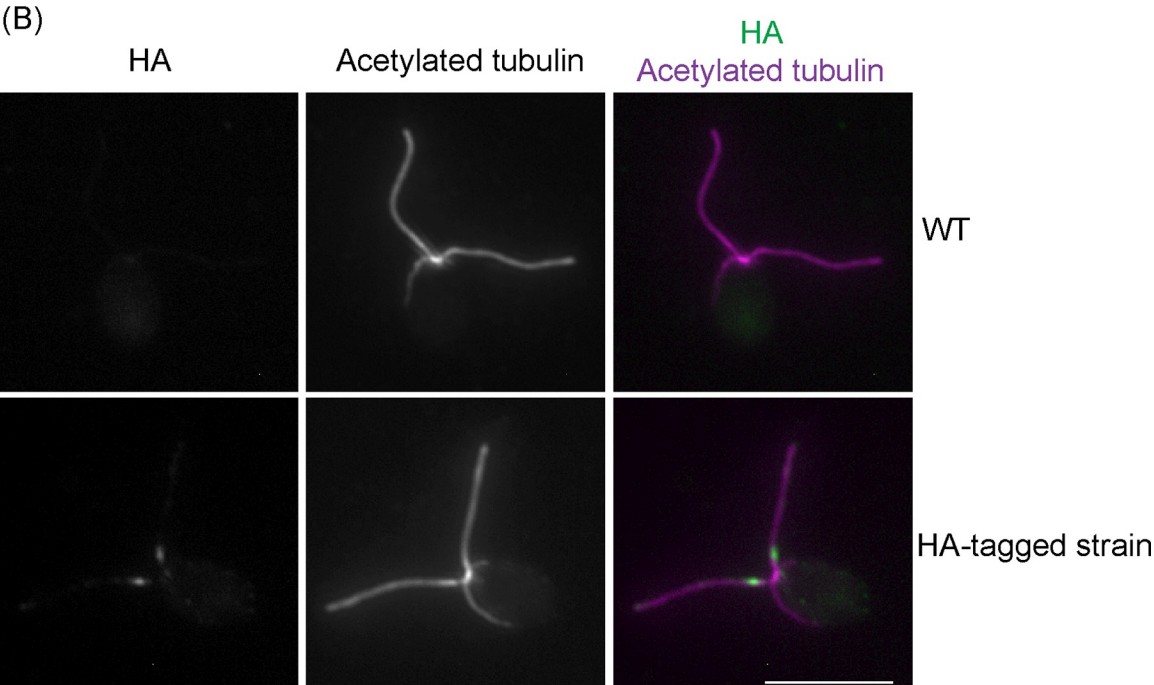

**Fig 6. LF5-HA protein is expressed and localized normally in 8 out of 10 PCR-positive clones.** (A) Representative western blots of whole-cell extracts of cells of wild-type strain g1 (WT), the *LF5*-null mutant *lf5-2*, and 10 positive clones from the initial PCR screen. The blots were probed with anti-LF5 antibody (top panel) or anti-HA antibody (middle panel). Eight out of 10 PCR-positive clones expressed near normal levels of LF5-HA. ATP synthase β subunit (βF1-ATPase) was probed as a loading control (lower panel). The HA-tagged LF5 migrates slightly slower than non-tagged LF5. In these blots of whole-cell extracts, the anti-LF5 antibody labels a broad, diffuse band in the LF5 region; this band is present in *lf5-2*, indicating that it is non-specific. (B) Representative immunofluorescence microscopy images of cells of the wild-type strain g1 (WT) and one of the HA-tagged strains. Cells were probed with anti-HA antibody (shown in green in merged images) and anti-acetylated tubulin antibody (shown in magenta in merged images). The HA signal is localized to the flagella with concentration at the proximal end of the flagellar shaft, which is the typical localization pattern for LF5 [14]. Scale bar: 10 µm.

(A)

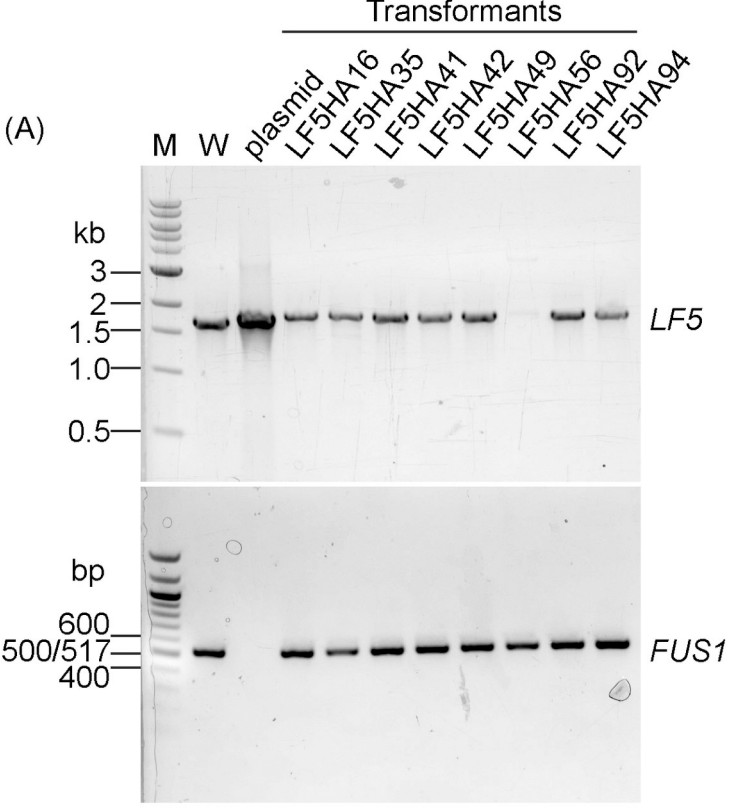

(B)

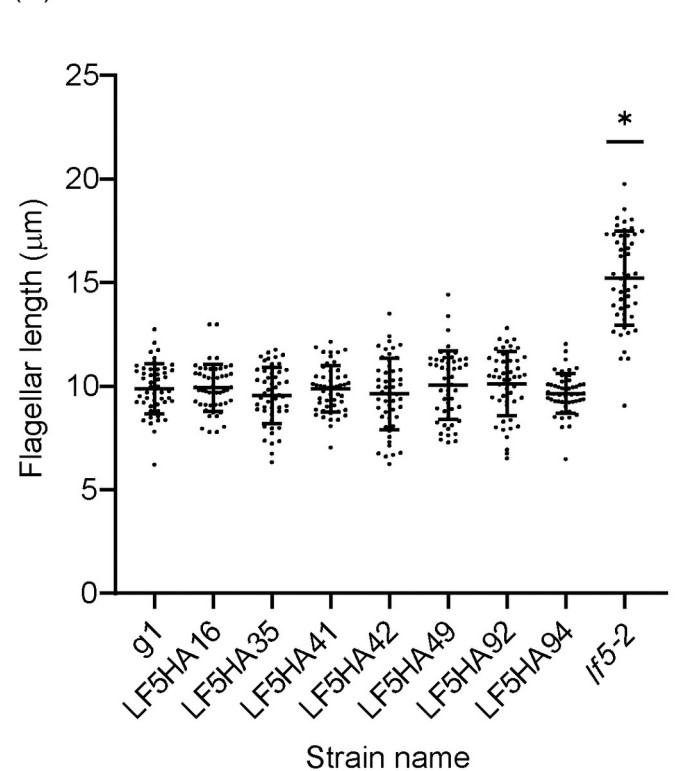

**Fig 7. Seven out of the eight strains that express LF5-HA have an intact *LF5* 3' UTR and have normal flagellar length.** (A) Agarose gels showing the PCR results to check the integrity of the 3' UTR on the eight strains that expressed near normal levels of LF5-HA. W: wild-type g1 genomic DNA as template. Plasmid: plasmid pLF5HA, which contains the *LF5-HA* gene, as control. Transformants: genomic DNA from the eight HA-tagged strains. Wild-type product (1674 bp) migrates slightly faster than product from the HA-tagged strains (1785 bp). Mating-type-plus-specific gene *FUS1* was used as a positive control. M: NEB 1 kb DNA ladder (catalog number NO552) on the upper gel and NEB 100 bp DNA ladder (catalog number NO551) on the lower gel. (B) Flagellar length is normal in the seven strains that express LF5-HA and have a complete *LF5* 3' UTR. g1 is the wild-type strain in which the editing experiment was done. When compared with g1, only the null mutant *lf5-2* shows significant difference (P<0.05) as marked by *.

either the 3' end of the donor DNA or the new 5' end generated by the DSB could be used as template.

To investigate if the gene was edited as expected at the RNP1 cut site, we carried out PCR analysis of the 96 clones using primer pair P3/P4 as illustrated in Fig 4. However, instead of getting a product of 542 bp as expected for a perfect edit, we observed larger products ranging from approximately 1000 bp to 1500 bp (Fig 9). Of the 16 clones that yielded product of the expected size when using primer pairs P1/P2 (Fig 8), only 10 yielded PCR product in this round. To understand the reason for these results, we sequenced the 10 PCR products (S4 Appendix) and found that RNP1 had not cut the target site in any of the clones, and that the donor DNA had inserted at the RNP2 cut site with modifications (indels) at its 5' end, resulting in duplication of sequence upstream of the cut site (Fig 10). Subsequently, we tested RNP1's ability to cut *NAP1L1* DNA *in vitro* and found that it can do so as effectively as RNP2 (S3 Fig). Moreover, RNP1 and RNP2 together cut *NAP1L1* DNA into three fragments of the expected sizes, indicating that the two can function together *in vitro*.

We carried out western blotting to determine if NAP1L1 protein with N-terminal mNeon-Green-3xFLAG tag was expressed in any of the 16 clones that initially yielded a PCR product of the expected size for correct insertion of the 3' end of the donor DNA at the RNP2 site. Probing with anti-FLAG antibody revealed that 10 of the clones did indeed express a FLAG-tagged protein, not present in wild type, of the expected mass for tagged NAP1L1 (Fig 11), strongly suggesting that the expressed protein is NAP1L1- mNeonGreen-3xFLAG. Nine of the 10 had a precise insertion at the 3' end of the donor DNA as shown in S3 Appendix. The tenth (clone F9, sequence SX3 in S3 Appendix) had a 36-bp deletion in the first intron. It is possible that shortening of this intron did not affect the splicing, so that a wild-type NAP1L1 with tag is still expressed. Different amounts of the tagged protein were expressed in the different clones, possibly reflecting different modifications to the regulatory sequences upstream of the *NAP1L1* coding region (see Discussion). Of the 16 clones that initially yielded a PCR product of the expected size for correct insertion of the 3' end of the donor DNA at the RNP2 site (Fig 8), 3 clones (B11, D3, and F11) have precise integration at the 3' end of the donor DNA (S3 Appendix) yet did not express the tagged proteins. The integration of the 5' end of the donor DNA in these clones apparently was accompanied by complicated events such as large insertions, deletions, or reorganizations, as evidenced by the absence of PCR products in Fig 9. In each case, the more complicated event presumably abolished expression of the tagged protein.

We were unable to detect mNeonGreen signal in living cells or anti-FLAG signal in fixed cells. Although the tagged protein can be detected by western blotting, it apparently is not expressed at a level sufficiently high to be detected at the single-cell level with our microscopes. Consequently, the location of NAP1L1 within the cell remains unknown.

Because editing of *NAP1L1* occurred as in Fig 10 instead of as envisioned in Fig 4, this experiment in effect confirmed that a single-RNP strategy (Fig 3) can be used to tag the N-terminus of a protein. The tagging efficiency for this experiment was 9% (9 out of 96 transformants).

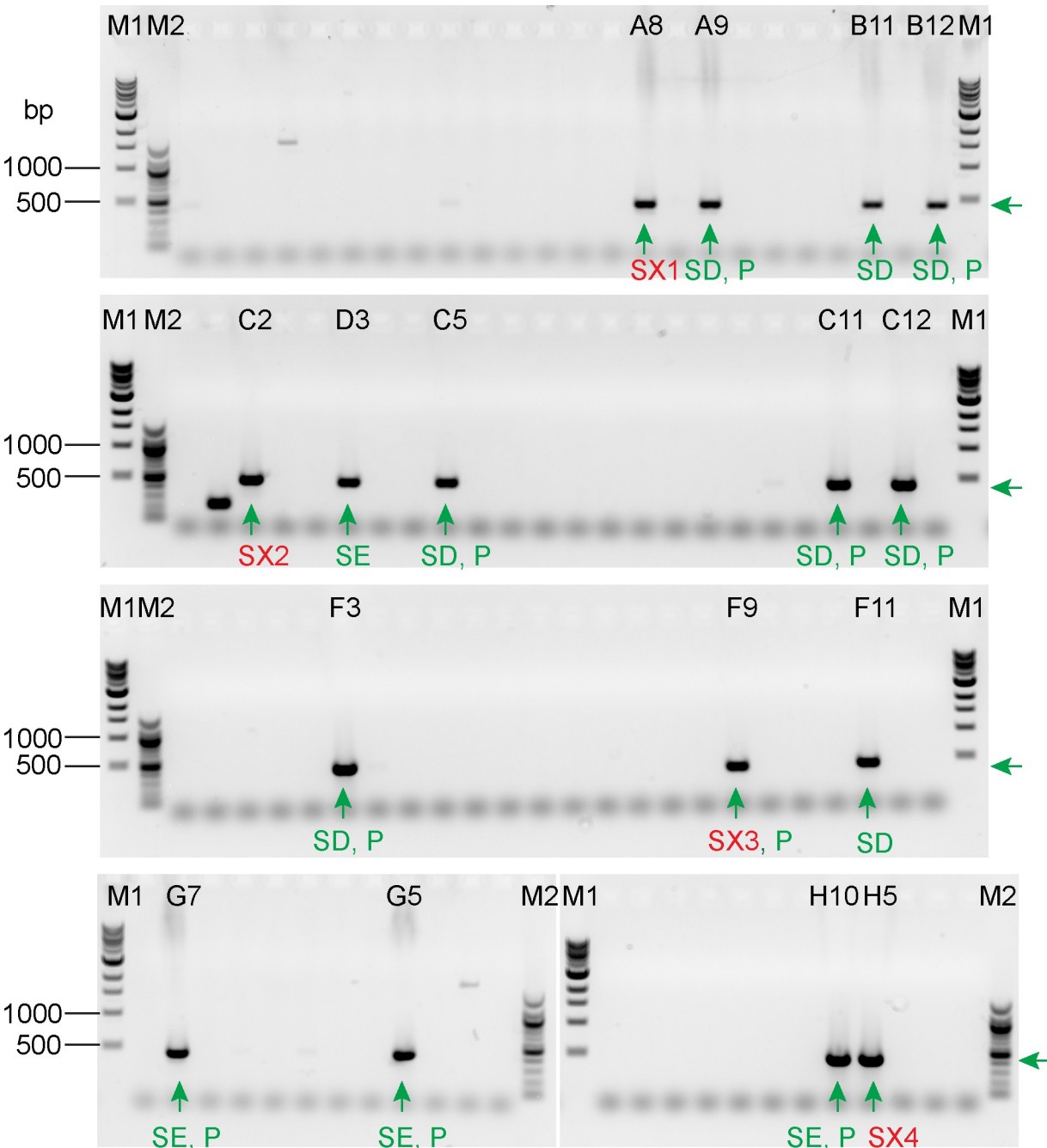

**Fig 8. PCR screening of transformants for insertion of the 3' end of the NAP1L1 mNeonGreen-3xFLAG donor DNA at the RNP2 cut site.**
Crude DNA preparations from 96 transformants were used as templates for PCR amplification with primer pair P1/P2 as illustrated in Fig 4. Since P1 is complementary to the tag sequence, no PCR product is expected for the wild-type gene. The expected size for product from a correctly tagged gene is 450 bp. The presence of product of a size other than 450 bp indicates that more complicated editing occurred. Sixteen clones (labeled A8-H10) yielded product of the size expected for a correct edit (green arrows below the bands and on the right side of the gels); the PCR products from these clones were sequenced for further confirmation (S3 Appendix). Eight of the 16 (marked with green SD) had a sequence that was exactly as expected if the donor DNA was used as the repair template, whereas four (marked with green SE) had a sequence that was exactly as expected if the endogenous DSB end was used as the repair template. In the remaining four (red SX1-SX4), indel or point mutations had been introduced. Ten clones subsequently found to express NAP1L1 tagged with mNeonGreen-3xFLAG (see Fig 11) are marked with a green P. M1: NEB 1 kb DNA ladder (catalog number NO552). M2: NEB 100 bp DNA ladder (catalog number NO551). The band sizes from top to bottom for M1 are 10, 8, 6, 5, 4, 3, 2, 1.5, 1, and 0.5 kb. The band sizes from top to bottom for M2 are 1517, 1200, 1000, 900, 800, 700, 600, 500/517, 400, 300, 200, and 100 bp.

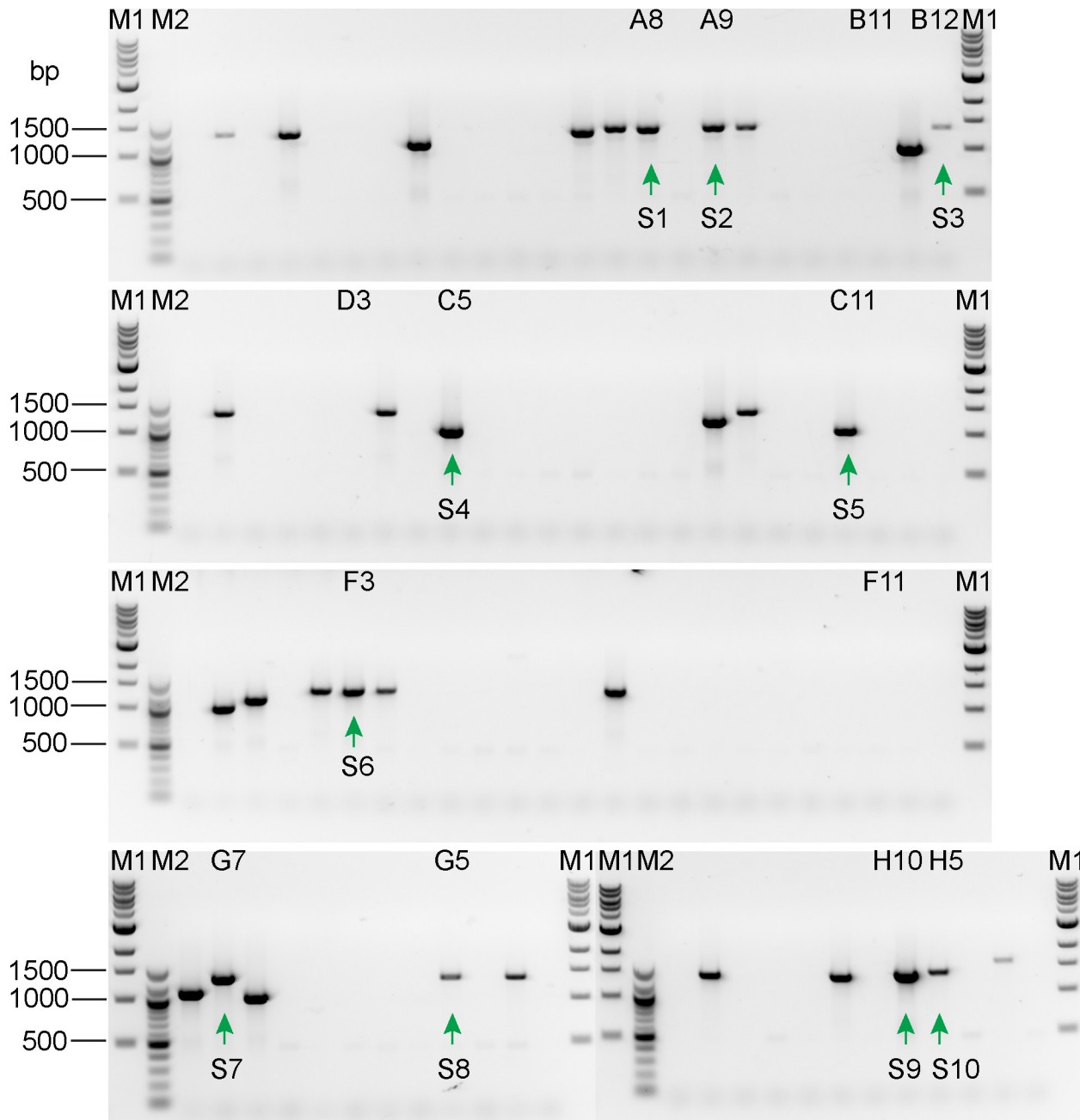

**Fig 9. PCR screening of transformants for insertion of the 5' end of the NAP1L1 mNeonGreen-3xFLAG donor DNA at the RNP1 cut site.**
Crude DNA preparations from the same 96 transformants and in the same order as those analyzed in Fig 8 were used as templates for PCR amplification with primer pairs P3/P4 as illustrated in Fig 4. Because P4 is complementary to sequence in the antibiotic-resistance gene, no PCR product is expected for the wild-type gene. The expected product size for correct editing at the RNP1 target site is 542 bp. However, the PCR products are around 1–1.5 kb. A product of 1350 bp is expected if the RNP1 target site was not cut and the donor DNA inserted cleanly at the RNP2 cut site. To explore this possibility, we sequenced product (here marked S1-S10 below the bands) from 10 clones (labeled at the top of the lanes as in Fig 8) that had yielded product of the correct size with primer pair P1/P2. The results (S4 Appendix) confirmed that the RNP1 target site was not cut and the donor DNA had inserted into the RNP2 cut site, but with modification of its 5' end. M1: NEB 1 kb DNA ladder (catalog number NO552). M2: NEB 100 bp DNA ladder (catalog number NO551). The band sizes from top to bottom for M1 are 10, 8, 6, 5, 4, 3, 2, 1.5, 1, and 0.5 kb. The band sizes from top to bottom for M2 are 1517, 1200, 1000, 900, 800, 700, 600, 500/517, 400, 300, 200, and 100 bp.

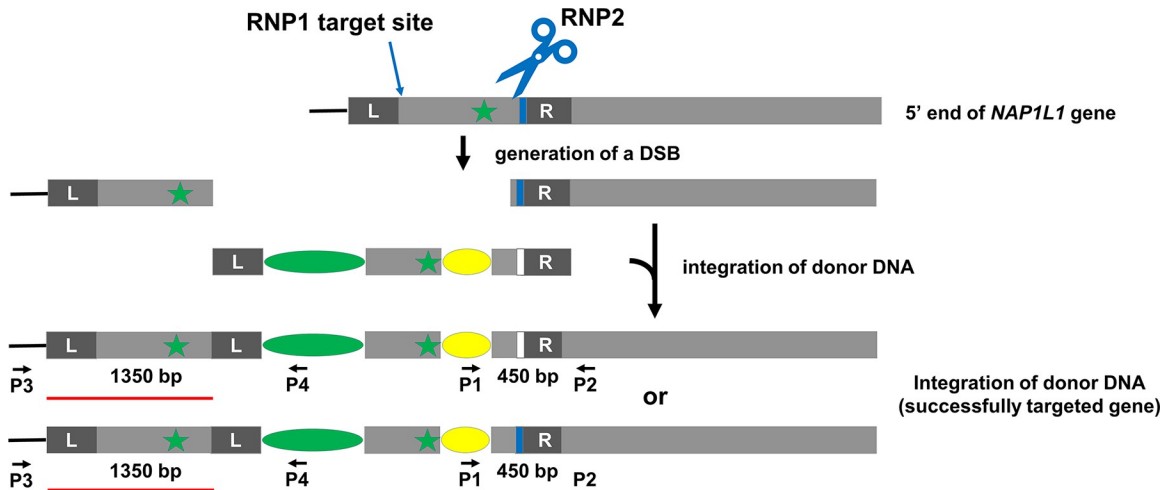

**Fig 10. Schematic overview of TIM-based N-terminal tagging of NAP1L1.** Two RNPs were used as described in Fig 4. However, the gene was cut by RNP2 only, resulting in an insertion of the donor DNA at that site. A portion (indicated by a red line) of the original gene remains immediately upstream of the drug-resistance cassette. The sizes of the expected PCR products for specific experiments of Figs 8 and 9 are shown between the primers.

## Discussion

*C. reinhardtii* has numerous properties that have made it an excellent model organism for studying cell organelles and cell processes. Because it is unicellular and grows well in liquid or agar-based media, strains can be handled and maintained using conventional microbiological techniques. It can be grown cheaply and easily in large quantities, and methods have been developed for isolation of many of its cell structures; as a result, it is excellent for biochemical studies. For example, its flagella can be easily detached and isolated, giving a one-step enrichment of about 1000X for flagellar components. Its cell morphology, which has clear polarity and is nearly invariant, makes it well suited to imaging using a wide variety of light and

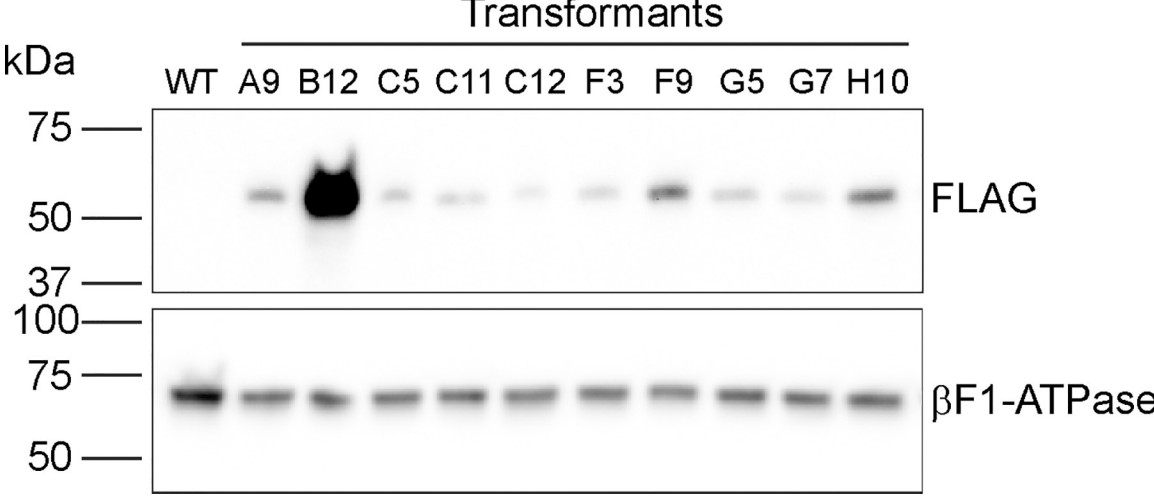

**Fig 11. NAP1L1-mNeonGreen-3xFLAG protein is expressed in 10 transformants.** Whole-cell lysates from the wild-type parental strain (WT) and 10 transformants (labeled as in Fig 8), each of which had a precise insertion at the 3' end of the donor DNA with the exception of clone F9, were probed with anti-FLAG antibody and anti-F1 beta ATPase antibodies for loading control. All transformants expressed FLAG-tagged protein at the expected mass (66 kDa) for NAP1L1-mNeonGreen-3xFLAG.

electron microscopy techniques. Because it has sex, it is ideal for genetic analyses. It is normally haploid, so even recessive mutations are immediately apparent; however, diploids can be formed, so mutants can be put into complementation groups. It also is excellent for molecular genetic approaches. Its nuclear genome, which has been sequenced, is readily transformed; as a result, it is possible to create insertional mutants, and it is possible to rescue mutants by transformation with a cloned gene. However, *C. reinhardtii* has lagged some other model organisms in the ease of making precise edits to the genome. Here we show that simple modifications of a CRISPR/Cas9-based method for targeted insertional mutagenesis [10] can be used to efficiently and specifically modify endogenous genes *in vivo* so that they express proteins that are tagged at either their N or C terminus but are otherwise wild type.

## Importance of additional characterization of edited strains identified in initial PCR screens

In our experiments, Cas9/gRNA RNP and donor DNA encoding both the desired tag sequence and sequence conferring antibiotic resistance were delivered into wild-type cells, which were then plated on antibiotic-containing agar. Individual colonies were then selected and screened by PCR analysis using primers that amplified the target site to identify those clones in which the gene may have been modified as intended. The PCR products of those transformants that passed this initial PCR screening were then sequenced to check the integrity of the 3' end of the gene in the case of C-terminal tagging and the 5' end of the gene in the case of N-terminal tagging. In addition, the strains were characterized at the protein and phenotype levels. Each of these additional checks is important before concluding that a strain is edited as desired, because, in addition to perfect insertions, an insertion may be accompanied by modification of the donor DNA ends, concatenation of the donor DNA, deletion of flanking genomic DNA, or reorganizations of the target site, all of which are common to insertional mutagenesis techniques [27], which is the basis of TIM-tagging.

Our initial screen for C-terminal tagging of LF5 identified 11 clones that yielded PCR product of the size expected for a correct insertion, but only seven of these passed the tests of DNA sequencing and of western blotting for tagged protein expression. All seven had wild-type phenotypes. For the N-terminal tagging of NAP1L1, among the 16 clones that were identified in our initial screen for PCR products of the size expected for correct insertion of the 3' end of the donor DNA, sequencing showed that 12 had correct insertions at the 3' end of the donor DNA. Of these 12, nine expressed FLAG-tagged protein. All nine appeared to have wild-type phenotypes.

In contrast to strains expressing LF5-HA, mNeonGreen-3xFLAG-tagged NAP1L1 strains expressed variable levels of the tagged protein. One strain, B12, had relatively high levels of expression. Sequence analysis showed that this strain has an indel mutation at the 5' end of the inserted donor DNA. In the case of a simple integration as illustrated in Fig 10, this indel is well upstream of the start codon of the tagged gene and therefore would not be expected to affect its coding region. However, during donor DNA integration, in addition to precise integration events, other events such as large deletions, insertions, duplications, and conversions may happen in some strains. We sequenced only two small regions surrounding the ends of the integrated donor DNA. It is possible that a more complicated event happened between these two regions with a resulting effect on the regulation of expression.

During transformation, insertions may occur in other places in the genome, and this may cause unexpected variation in phenotype. This is a common concern for all transformation experiments using a selectable marker. There are ways–e.g., whole-genome sequencing or Southern blotting—to determine if the resistance gene has inserted only in the gap created by

the RNP and not elsewhere. However, in the case of TIM tagging, we were easily able to obtain multiple tagged strains, so it was more practical to study the different tagged strains to determine if they retained a wild-type phenotype and if the tagged protein behaved the same in the different strains. A strain that does not pass these checks should be suspect, and therefore be discarded or crossed to a wild-type strain to determine if the unexpected phenotype segregated from the tagged gene.

## Homology arms are critical

Unlike the TIM method for creating insertional mutants, homology arms are critical for this tagging strategy. Homology arms not only increase the efficiency of insertion of donor DNA [10], but also promote accurate repair at the ends of the inserted DNA. This is evident here in both the LF5- and NAP1L1-tagging experiments. In the case of LF5 tagging, the donor DNA had a single "left" homology arm matching sequence immediately upstream of the RNP cut site, and 10 of 11 clones that yielded PCR product of the expected size for a correct insertion were perfectly edited at this end of the inserted DNA (Fig 5 and S1 Appendix). In contrast, none of the 7 clones sequenced had a perfect insertion at the 3' end of the donor DNA, where there was no homology arm (S2 Appendix). In the case of NAP1L1 tagging, the 3' end of the donor DNA had a "right" homology arm that matched sequence just downstream of the RNP2 cut site, and this end of the donor DNA was precisely inserted in 12 out of 16 clones that had yielded PCR product of the expected size (Fig 8 and S3 Appendix). However, because RNP1 did not cut the gene, with the result that both ends of the donor DNA were inserted at the RNP2 cut site, the "left" homology arm on the donor DNA did not match sequence immediately upstream of the cut site, and none of the sequenced clones were perfectly edited at the 5' end of the donor DNA (Fig 9 and S4 Appendix).

In our experimental design for tagging the C-terminus of a protein (Fig 2), we did not include a right homology arm because this end of the donor DNA, once integrated, will be downstream of the final tagged gene and the drug-resistance gene, where a precise integration is not required. Similarly, in our strategy for using a single RNP to tag the N-terminus of a protein (Fig 3), we did not include a left homology arm because this end of the donor DNA, once integrated, will be upstream of the drug-resistance gene and the final tagged gene. However, in our strategy using two RNPs to tag a protein (Fig 4), we wished to have precise HDR at both ends of the inserted DNA. Therefore, we included a homology arm on each end of the donor DNA. Integration of both ends of the donor DNA by HDR likely would reduce the chance of indel mutations, large deletions, rearrangements and so on that would affect expression of the gene. The use of both left and right homology arms, rather than just a single homology arm, also may decrease the number of off-target or missed-target events.

## TIM-tagging is flexible

TIM-tagging can be modified to suit various needs. First, different selectable markers can be used. We used the *Streptomyces rimosus aphVIII* gene conferring resistance to paromomycin, and the *Streptomyces hygroscopicus aph7"* gene conferring resistance to hygromycin. The ability to use different selectable markers is an advantage if one wishes to tag a protein in a strain already carrying a selectable marker. Second, different tags can be used. We used HA, which is ideal for western blotting, immunofluorescence microscopy and immunoprecipitation of tagged proteins in *C. reinhardtii*, and mNeonGreen-3xFLAG, which is not only suitable for western blotting and immunoprecipitation but also enables the dynamic movement of a protein to be followed *in vivo* using advanced microscopy methods. Third, the strategy can be applied to different strains, with no obvious requirements for strain background except that

the strain be amenable to transformation. We used strain g1, a wild-type strain that has a cell wall and is readily transformed [28]. The applicability of the strategy to a variety of strains would be of utility if one wished to tag gene A in a gene B mutant background. Conventionally, this was done by making or acquiring two strains: one that expresses tagged transgene A in a gene A null background, and a second that has a mutation in gene B. Then the two strains would be crossed, and the offspring would be screened for those that carry the tagged trans-gene A, the endogenous null gene A, and the mutant gene B. Our TIM-tagging strategy could be used to replace this time-consuming and complicated process by transforming the gene B mutant directly with RNP and donor DNA targeting gene A. The screening process would remain the same as that used here for wild-type cells.

## One-RNP strategy vs. two-RNP strategy

Using a one-RNP strategy results in the duplication of genomic DNA, either upstream of the insertion site in the case of N-terminal tagging (Fig 3), or downstream of the insertion site in the case of C-terminal tagging (Fig 2). In the latter case, the duplication of sequence is unlikely to affect expression or function of the gene product, because it includes only a small amount of 3'-coding sequence plus the remaining 3' end of the gene, and is downstream of the tagged gene and the drug-resistance gene. However, in the former case, duplication of sequence that is upstream of the target site could affect gene regulation because this is where the *cis*-acting transcriptional regulatory DNA elements are located. Indeed, in our tagging of NAP1L1, in which the 5' end of the gene was duplicated as a result of failure of RNP1 to cut its target *in vivo*, we observed more clonal variation in the expression level of the tagged protein than was the case for tagged *LF5*, in which the duplicated sequence included only the 3' end of the gene and was downstream of the gene. Therefore, we believe that the two-RNP strategy is more appropriate for tagging the N-terminus of a protein. The two-RNP strategy also could be used for tagging the C terminus of a protein in cases where one wished to avoid duplication of sequence downstream of the gene.

It is unclear why RNP1 failed to cut its target site in our test of the two-RNP strategy. The use of two gRNAs to edit a gene works in other systems [29–32], and it works on the *C. reinhardtii* chloroplast genome [33]. In two of these studies, the target cut sites were separated by 960 bp [31] and 32 to 98 bp [32]; for comparison, in our experiment the two target sites were separated by 767 bp, so the linear distance between the sites is unlikely to have hindered cutting by RNP1. Indeed, *in vitro* experiments showed that RNP1 and RNP2 together cut both target sites in *NAP1L1* DNA substrate efficiently and completely, so the failure of RNP1 is unlikely to have been due to a poorly designed gRNA. It is possible that RNP1 could not access the cutting site *in vivo* due to regulatory proteins bound there. Further experiments will be necessary to determine if this is a result specific to *NAP1L1*, or a more general condition and, if the latter, how to address the issue.

## Prospects for use of Cas9 variants in TIM-tagging

We have used wild-type spCas9 for our TIM-tagging. Ideally, in the case of the one-RNP approach, the PAM sequence for C- or N-terminal tagging would be located immediately in front of the stop codon or immediately after the start codon, respectively, to minimize the size of the duplicated region in the final allele. In the case of the two-RNP approach, the PAM sequence that is upstream of the gene for N-terminal tagging or downstream of the gene for C-terminal tagging ideally would be located between the targeted gene and the neighboring gene to avoid mutating the latter. In situations where there may not be a Cas9 PAM site at the ideal location, near-PAMless engineered CRISPR-Cas9 variants [34] or different Cas endonucleases

that recognize other PAMs [35] might enable access to these sites. Thus, we do not anticipate that lack of appropriately located PAM sequences will limit the utility of this method for tagging other nuclear genes.

### TIM-tagging should enable precise editing within a gene

While we focused on tagging genes at their ends, theoretically, our TIM-tagging strategy also could be used to introduce a site-specific mutation into a gene, or to place a tag in the interior of the gene. The former should be accomplished quite readily using one of our two strategies if the sequence being edited is near the 3' or 5' end of the gene. In this case the donor DNA would contain a portion of the gene's sequence, encoding the desired mutation and extending from the region to be edited to the 3' or 5' end of the gene (whichever is closer), followed or preceded by the selectable marker, respectively. Edits further from an end of the gene would require a longer donor DNA, which would be expected to decrease both the efficiency of transformation and the likelihood of perfect insertions. For tagging of LF5, we used a 3.8-kb donor DNA and achieved an efficiency of 7.4%, while for NAP1L1 we used a 3.3 kb donor DNA and observed an efficiency of 9%. In cases where the efficiency of a perfect insertion may be lower, e.g., when using a two-RNP strategy or using a much longer donor DNA, DNA from multiple colonies could be pooled for the initial PCR-based screen, and colonies that went into positive pools could then be tested individually to identify positive colonies.

### Conclusions

Using this tagging strategy derived from TIM, we successfully tagged LF5 with HA at the C-terminus with an efficiency rate of 7.4% and NAP1L1 with mNeonGreen-3xFLAG at the N-terminus with an efficiency rate of 9%. The strategy is flexible in that there is no requirement to start with a specific strain, and the approach can accommodate different tags and different selectable markers. Appropriate cut sites for wild-type SpCas9, which was used in our study, are common in the *C. reinhardtii* genome, but Cas9 variants and different endonucleases that recognize other PAMs are available if wild-type SpCas9 PAM sequences are not present at the desired target site. TIM-tagging is relatively simple and can be carried out more quickly than the current conventional approach because it does not require the construction of a null mutant for the gene of interest. Therefore, we expect that this method will be equally useful for tagging most, if not all, other nuclear genes, including essential genes. Further refinement of this approach may enable precise deletion, substitution, or addition of specific nucleotides or short segments of sequence within a gene.

### Supporting information

**S1 Appendix. Sequences of PCR products for 1) initial screening of LF5-HA strains (related to Fig 5 of main text) and 2) to check integrity of 3' UTR of LF5-HA strains (related to Fig 7 of main text).**
(DOCX)

**S2 Appendix. Sequences of PCR products amplified from the region surrounding the 3' end of the inserted donor DNA in LF5-HA strains (related to Figs 6 and 7 of main text).**
(DOCX)

**S3 Appendix. Sequences of the PCR products for initial screening of NAP1L1-tagged strains (related to Fig 8 of main text).**
(DOCX)

**S4 Appendix. Sequences of the PCR products amplified from the region surrounding the 5' end of the inserted donor DNA in NAP1L1-tagged strains (related to Fig 9 of main text).** (DOCX)

**S5 Appendix. Plasmid sequences.** (DOCX)

**S1 Table. List of gRNAs and primers.** (DOCX)

**S1 Fig. Schematic representation of vector construction.** (A) Schematic diagram illustrating how vector pLF5CsfGFP was generated. Fragments 1 and 2 were amplified from the plasmid pBS3830, which contains the *LF5* gene. Fragment 1 contains the 5' end of the *LF5* gene up to the stop codon while fragment 2 contains the 3' end of the *LF5* gene starting with the stop codon. Fragment 3, which contains the coding region for sfGFP, was amplified from the plasmid pIFT140-sfGFP-aphVIII. Fragment 4, containing the paromomycin cassette, was created by linearizing plasmid pKS-aphVIII-lox with SacI. The four fragments were ligated together using NEBuilder® HiFi DNA Assembly Master Mix to produce the vector pLF5CsfGFP. The *LF5* gene is indicated by a wide grey line, sfGFP is indicated by a wide orange line, and the paromomycin cassette is indicated by a wide green line. The primers are shown above the corresponding amplification regions as short arrows with thickness and color corresponding to the plasmid sequence that they match. The red star represents the stop codon of the *LF5* gene. (B) Schematic representation of how vector pLF5HA was made. All of plasmid pLF5CsfGFP except for the sfGFP sequence was amplified by PCR to generate Fragment pLF5CsfGFPΔsfGFP. The 3HA sequence was cut from plasmid p3HA. These two fragments were ligated using NEBuilder® HiFi DNA Assembly Master Mix to produce the vector pLF5HA. The *LF5* gene is indicated by a wide grey line, sfGFP is indicated by a wide orange line, the paromomycin cassette is indicated by a wide green line, and sequence encoding the 3HA tag is indicated by a wide blue line. The primers are shown below the corresponding amplification regions as short arrows with thickness and color corresponding to the plasmid sequence that they match. The red star represents the stop codon of the *LF5* gene. (PDF)

**S2 Fig. PCR analysis of transformants that yielded no band during initial PCR screen for LF5-tagged strains confirmed that the *LF5* locus was disrupted.** To confirm that the lack of PCR products was caused by disruption of the *LF5* gene, genomic DNA from strains that did not yield any product in the initial PCR screening for LF5HA strains (Fig 5) were amplified using primer pairs specific to the *LF5* locus and to mating-type-plus-specific gene *FUS1* as positive control. While primers specific for *FUS1* yielded products for all the transformants, there was no amplification using primers specific for *LF5*, indicating that the *LF5* locus was disrupted in these transformants. W: wild-type strain g1 as control; HA: plasmid containing *LF5HA* as template; M: NEB 100 bp DNA ladder (catalog number N0551). (DOCX)

**S3 Fig. Both NAP1L1 RNP1 and RNP2 cut *NAP1L1* DNA targets *in vitro*.** *NAP1L1* DNA was amplified and digested *in vitro* as described in the Materials and Methods, and the digestion products were separated on a 1.2% agarose gel. The *NAP1L1* DNA substrate is 2.8 kb in size; RNP1 was predicted to yield fragments of 1322 bp and 1478 bp, while RNP2 was predicted to yield fragments of 2109 bp and 691 bp. In the experiment, each individual RNP completely digested the substrate, yielding products of the expected sizes. RNP1 and RNP2 together completely digested the substrate into three fragments of 1322 bp, 787 bp, and 691 bp,

indicating that the two RNPs can cut the same fragment *in vitro*. C: *NAP1L1* DNA only. M1: 1 kb DNA ladder (NEB catalog #N0552). M2: 100 bp DNA ladder (NEB catalog #N0551); from top to bottom, markers are 1517, 1200, 1000, 900, 800, 700, 600, 500/517, 400, 300, 200, and 100 bp.
(DOCX)

**S1 Raw images. Original western blot and agarose gel images.**
(PDF)

## Acknowledgments

We thank Dr. Paul A. Lefebvre (University of Minnesota) for the generous gifts of the antibody to *C. reinhardtii* LF5 and the plasmid pBS3830 that contains the wild-type *LF5* gene. We are grateful to Drs. Michael Stuck and Greg Pazour (University of Massachusetts Chan Medical School) for help with fluorescence microscopy.

## Author Contributions

**Conceptualization:** Yuqing Hou, George B. Witman.

**Funding acquisition:** Yuqing Hou, George B. Witman.

**Investigation:** Yuqing Hou, Xi Cheng.

**Project administration:** Yuqing Hou, George B. Witman.

**Supervision:** Yuqing Hou, George B. Witman.

**Visualization:** Yuqing Hou, George B. Witman.

**Writing – original draft:** Yuqing Hou, George B. Witman.

**Writing – review & editing:** Yuqing Hou, George B. Witman.

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
