## [Decision Letter · Decision Letter 0]

16 Aug 2022

PONE-D-22-20272Direct in situ protein tagging in Chlamydomonas reinhardtii utilizing TIM, a method for CRISPR/Cas9-based targeted insertional mutagenesisPLOS ONE

Dear Dr. Witman,

Thank you for submitting your manuscript to PLOS ONE. After careful consideration, we feel that it has merit but does not fully meet PLOS ONE’s publication criteria as it currently stands. Therefore, we invite you to submit a revised version of the manuscript that addresses the points raised during the review process. Your manuscript was reviewed by four experts. While they found it interesting, they also raised some concerns. Please revise it according to their suggestions. Please note that some of them submitted their comments with attachments. Please also respond to them.

We look forward to receiving your revised manuscript.

Kind regards,

Hodaka Fujii, M.D., Ph.D.

Academic Editor

PLOS ONE

Journal Requirements:

2. We noted in your submission details that a portion of your manuscript may have been presented or published elsewhere. 

"Fig 1 was modified from Fig 1A from our paper "TIM, a targeted insertional mutagenesis method utilizing CRISPR/Cas9 in Chlamydomonas reinhardtii", which was published in 2020 in PLoS One. This figure is used in the introduction of this manuscript to give readers the necessary background information." 

Reviewers' comments:

Reviewer's Responses to Questions

**Comments to the Author**

1. Is the manuscript technically sound, and do the data support the conclusions?

Reviewer #1: Yes

Reviewer #2: Yes

Reviewer #3: Yes

Reviewer #4: Yes

2. Has the statistical analysis been performed appropriately and rigorously? 

Reviewer #1: Yes

Reviewer #2: Yes

Reviewer #3: Yes

Reviewer #4: N/A

3. Have the authors made all data underlying the findings in their manuscript fully available?

Reviewer #1: Yes

Reviewer #2: Yes

Reviewer #3: Yes

Reviewer #4: Yes

4. Is the manuscript presented in an intelligible fashion and written in standard English?

Reviewer #1: Yes

Reviewer #2: Yes

Reviewer #3: Yes

Reviewer #4: Yes

5. Review Comments to the Author

Reviewer #1: The manuscript by Hou et al titled “Direct in situ protein tagging in Chlamydomonas reinhardtii utilizing TIM, a method for CRISPR/Cas9-based targeted insertional mutagenesis” reported that they successfully tagged the C-terminus of wild type LF5 with the hemagglutinin (HA) tag with an efficiency of 7.4%, and the N-terminus of Cre09.g416350 with mNeonGreen-3xFLAG with an efficiency of 9%. Furthermore, they showed that the tag region in these strains is as expected, and confirmed the expression of tagged protein of the expected size in these strains. This is the first time that C. reinhardtii endogenous genes have been edited in situ to express a wild-type tagged protein, and TIM-tagging strategy promises to be a useful tool for the study of nuclear genes including essential genes in C. reinhardtii.

This well-written manuscript, is basically suitable for publication in PLOS ONE. Several concerns however are found, as described below.

1. The authors stated that homology arms are critical is this strategy developed in this manuscript. However, only left arm is included in TIM-tagging LF5 with HA at the C terminus. How the C-terminal of the antibiotic cassette (e.g., AphVIII, green oval in Figure2) was integrated into the sequences of the gene of interest? Is it by non-homologous end joining (NHEJ)? Why right left arm is not considered here? and the same case for the strategy shown in Figure3. However, both arms (left and right) are used in the strategy shown in Figure 4?

2. In the strategy shown in Figure3, it might be better if the both arms (left and right) are used to avoid off-target or miss-target?

3. In line 240, it is hard to understand about the sentence “The region between the cut site and the end of the gene is duplicated downstream of the drug-resistance cassette (dashed line with arrows on both ends)”. The authors should make it more clear especially about the meaning of “dashed line with arrows on both ends” in the figure.

4. The authors emphasized that “Importance of additional characterization of edited strains identified in initial PCR screens”. However, the localization of NAP1L1-mNeonGreen-3xFLAG was missed. Is it correctly target to the predicted organelle?

5. During the delivery of Cas9/gRNA RNP and donor DNA, why “The cuvette was then

incubated immediately at 16°C for 1 h”？Is this more efficient than conventional method （e.g., RT for 10 min）？

6. In line 114 “plasmid pLF5CsfGFP was first constructed by piecing together four fragments using NEBuilder® HiFi DNA Assembly Master Mix (catalog number E2621)”. The readers do not have idea how the author constructed the vectors. Please make it clearer that why these four fragments can be assembled when the authors designed the primers to amplify each fragment? It is better to make schematic representation as one of the supplementary figures.

7. It is likely that Figure 1 is as same as the image appeared in PLoS One. 2020 May 13;15(5):e0232594. For the schematic representation data, such Figure 1-4 should move to supplementary figures.

Reviewer #2: When submitting a paper to a journal, it is recommended that you standardize the format of the document according to the regulations of the journal, and edit the document within the level allowed by the journal so that reviewers can understand your research results more easily. Please next time should be more clear.

Reviewer #3: The authors performed endogenous gene editing of Chlamy. After RNP cleavage, the donor DNA (including homologous sequences, antibiotic resistance genes, tag genes, etc.) is introduced through homologous recombination, and the endogenous genes are tagged. In this paper, two genes were studied, one is LF5, which controls the length of the flagella, and tags HA at its C-terminus, and the other is NAP1L1, whose function is unknown, tags FLAG at its N-terminus.

All studies are based on the team's previous TIM (targeted insertional mutagenesis) design. The idea is: use RNP to cut the endogenous gene and introduce the donor DNA containing the antibiotic resistance gene (there is a homologous recombination arm on the left and right sides). The cutting pad is inserted through the homologous recombination arm.Two primers P1/P2 were designed at the endogenous gene cleavage site to identify the positive transformants. If the cleavage site is not inserted, the amplified fragment is a small fragment containing only part of the endogenous gene; if the cleavage site is inserted, the amplified fragment contains the endogenous gene and the resistance gene, and is a large fragment.

question:

1. In Figure 5, the authors performed sequencing analysis on 6 transformed strains (L1-L6) that amplified long fragments. The text shows that the sequence contains resistance genes, but the resistance genes are not marked in the S1 Appendix of the sequencing results. In theory, primers P1/P2 should amplify 563bp small fragments and large fragments in one reaction, but there is no small fragment and large fragment coexisting in one lane in the figure, how to explain?

2. In Figure 9, 10 NAP1L1 transformed strains expressed FLAG-tagged proteins. Question: B12 is of S3 type (deletion mutation and point mutation), but its FLAG expression is very high, why? Why do some S1 and S2 not express tagged proteins? All have been sequenced, and they should all be expressed.

3. S1-S10 marked in Figure 9 are easily confused with S1-S6 in Figure 8, so it is better to change the marking method.

Overall, the study is relatively complete. However, since it is the establishment of the methodology, the analysis of the edited gene sequences is insufficient, and some sequences are not annotated. It is possible that the authors focused on the expression of tagged proteins, thus weakening the sequence changes after gene editing. Another important issue is that many of the resistant transformants in this study did not undergo gene editing as expected, and the resistance gene was inserted elsewhere. So, is there some way to determine that resistance genes etc are inserted only in the gap created by the RNP and not elsewhere. Otherwise, there may be unexpected variation in phenotype if mutations are made in this way.

This manuscript is a further extension of the article published by their research group in Plos one in 2020. It is innovative and recommended to be published after revised.

Reviewer #4: The manuscript titled, ‘Direct in situ protein tagging in Chlamydomonas reinhardtii utilizing TIM, a method for CRISPR/Cas9-based targeted insertional mutagenesis’, by Hou et al has ued the green chlorophyte Chlamydomonas reinhardtii, a rather popular model system for flagellar studies to tag endogenous genes in situ using the modified TIM strategy. Undoubtedly, they have shown that this technique does circumvent the disadvantages of conventional approaches, the latter being more random. The authors have used this strategy on two genes: LF5/CDKL5, and Cre09.g416350/NAP1L1, both have not been studied previously in C. reinhardtii. The correct editing of these genes exhibit the success of the strategy (albeit at lower efficiencies).

I have only the following comments to make:

How equally useful is this strategy for other nuclear genes in C. reinhardtii?

What about the sfGFP in the final construct (lines 122-124)? Is it cleaved off?

Although the writing is simple and lucid, the authors have used three strains and must make mention of the type of strain they use in the M/M or wherever possible. It can get confusing, at times.

The authors have defended their modified TIM-based strategy to be relatively simple and quick, as opposed to the conventional approaches. A comparison in terms of time and ‘simplicity’ (whatever that implies) is required – especially since they claim this to be done for the first time in Chlamydomonas.

6. PLOS authors have the option to publish the peer review history of their article (what does this mean?). If published, this will include your full peer review and any attached files.

Reviewer #1: No

Reviewer #2: **Yes: **Jongrae Kim

Reviewer #3: No

Reviewer #4: No

---

## [Author Response · Author response to Decision Letter 0]

6 Nov 2022

Authors’ replies to specific reviewer and editor comments start with '***' below. Line numbers cited are for the revised version of the manuscript ‘Revised Manuscript with Track Changes’ when showing ‘All Markup.’

Dear Dr. Fujii and Reviewers:

We are grateful for the thoughtful comments of all four reviewers, and for their many positive remarks regarding our manuscript. We have accepted all suggestions and modified the manuscript accordingly. The original manuscript was focused on our successful application of the TIM-tagging approach; the revised manuscript retains this focus but now includes more sequence analysis and improved annotation of the Appendixes that will be helpful to those readers wishing to know more about the mechanism of donor DNA insertion in our experiments. We replaced Fig 1 with a new figure in the Introduction to avoid using a previously published figure and to better illustrate the background of this research and the TIM-tagging strategy. We also have addressed all the other questions asked by the reviewers, and made minor edits to improve syntax and clarity. We feel that the manuscript is substantially improved as a result. Specific replies to each point raised by the reviewers are below. 

Sincerely,

George Witman

Journal Requirements:

***The manuscript is now formatted according to PLOS ONE’s style requirements, including those for file naming.

2. We noted in your submission details that a portion of your manuscript may have been presented or published elsewhere. 

"Fig 1 was modified from Fig 1A from our paper "TIM, a targeted insertional mutagenesis method utilizing CRISPR/Cas9 in Chlamydomonas reinhardtii", which was published in 2020 in PLoS One. This figure is used in the introduction of this manuscript to give readers the necessary background information." 

***Fig 1 in the original submission was peer-reviewed and formally published in PLOS ONE. It was included to provide the reader with necessary background information. However, acting on the reviewers’ suggestions, we have now replaced Fig 1 with a new, previously unpublished, illustration better suited for this manuscript. Relevant wording in the Introduction has been revised accordingly.

***A new document named “S1_raw_images” is submitted in Supporting Information. This document contains all the original uncropped and unadjusted images underlying all blots and gel results reported in figures and supporting information files.

Reviewers' comments:

Reviewer's Responses to Questions

Comments to the Author

1. Is the manuscript technically sound, and do the data support the conclusions?

Reviewer #1: Yes

Reviewer #2: Yes

Reviewer #3: Yes

Reviewer #4: Yes

2. Has the statistical analysis been performed appropriately and rigorously? 

Reviewer #1: Yes

Reviewer #2: Yes

Reviewer #3: Yes

Reviewer #4: N/A

3. Have the authors made all data underlying the findings in their manuscript fully available?

Reviewer #1: Yes

Reviewer #2: Yes

Reviewer #3: Yes

Reviewer #4: Yes

4. Is the manuscript presented in an intelligible fashion and written in standard English?

Reviewer #1: Yes

Reviewer #2: Yes

Reviewer #3: Yes

Reviewer #4: Yes

5. Review Comments to the Author

Reviewer #1: The manuscript by Hou et al titled “Direct in situ protein tagging in Chlamydomonas reinhardtii utilizing TIM, a method for CRISPR/Cas9-based targeted insertional mutagenesis” reported that they successfully tagged the C-terminus of wild type LF5 with the hemagglutinin (HA) tag with an efficiency of 7.4%, and the N-terminus of Cre09.g416350 with mNeonGreen-3xFLAG with an efficiency of 9%. Furthermore, they showed that the tag region in these strains is as expected, and confirmed the expression of tagged protein of the expected size in these strains. This is the first time that C. reinhardtii endogenous genes have been edited in situ to express a wild-type tagged protein, and TIM-tagging strategy promises to be a useful tool for the study of nuclear genes including essential genes in C. reinhardtii.

This well-written manuscript, is basically suitable for publication in PLOS ONE. Several concerns however are found, as described below.

1. The authors stated that homology arms are critical is this strategy developed in this manuscript. However, only left arm is included in TIM-tagging LF5 with HA at the C terminus. How the C-terminal of the antibiotic cassette (e.g., AphVIII, green oval in Figure2) was integrated into the sequences of the gene of interest? Is it by non-homologous end joining (NHEJ)? Why right left arm is not considered here? and the same case for the strategy shown in Figure3. However, both arms (left and right) are used in the strategy shown in Figure 4?

***We thank the reviewer for bringing this up. We have now amplified the region surrounding the integration site of the 3’ end of the antibiotic-resistance cassette from the eight strains that express HA-tagged LF5 protein. PCR and sequencing results (new S2 Appendix) showed that three of them have in-del mutations at the integration site, which is typical for non-homologous end joining (NHEJ). The other strains have large insertions, deletions, and reorganizations, which is common for insertional mutagenesis in C. reinhardtii. Beginning on l. 507, we now briefly report our findings with regard to how the 3’ end of the antibiotic-resistance cassette was integrated into the genome.

In our experimental design for tagging the C-terminus of LF5 (Fig 2), we did not include a right homology arm because the 3’ end of the antibiotic-resistance cassette will be downstream of the final tagged gene and the drug-resistance gene, so a precise integration is not required. Similarly, we did not include a left homology arm in the strategy illustrated in Figure 3 because the 5’ end of the antibiotic-resistance cassette will be far upstream of the final tagged gene. In Fig 4, because we are using two gRNAs and wish to have precise integration at both ends of the donor DNA, the strategy includes both left and right homology arms. Beginning on l. 712, we now briefly explain why we did not include a right or left homology arm in the strategies diagrammed in Fig 2 and 3, respectively.

2. In the strategy shown in Figure3, it might be better if the both arms (left and right) are used to avoid off-target or miss-target?

***As discussed in the original version of our manuscript, we fully agree that two homology arms are preferable to one, especially when tagging the N-terminus of a protein. We now point out that the use of both left and right homology arms, rather than just a single homology arm, may decrease off-target or missed-target events (Discussion section ll. 722-723.)

3. In line 240, it is hard to understand about the sentence “The region between the cut site and the end of the gene is duplicated downstream of the drug-resistance cassette (dashed line with arrows on both ends)”. The authors should make it more clear especially about the meaning of “dashed line with arrows on both ends” in the figure.

***We thank the reviewer for pointing out that our sentence and diagram were confusing. In Figure 2, we have now changed the dashed line with arrows on both ends to a solid red line, and in the figure legend we have changed the sentence to “A portion (indicated by a red line) of the original gene remains immediately downstream of the drug-resistance cassette.” To make all figures consistent, the dashed lines in Figures 3 and 10 also have been changed to solid red lines, and the sentences mentioning these have been revised accordingly.

4. The authors emphasized that “Importance of additional characterization of edited strains identified in initial PCR screens”. However, the localization of NAP1L1-mNeonGreen-3xFLAG was missed. Is it correctly target to the predicted organelle?

***We have been unable to detect mNeonGreen signal in living cells or FLAG signal in fixed cells, and so could not determine the location of NAP1L1 within the cell. Although the tagged protein can be detected by western blotting, it apparently is not expressed at a level sufficiently high to be detected at the single-cell level with our microscopes. We have now added a short paragraph to the Results section to clarify this (ll. 613-616).

5. During the delivery of Cas9/gRNA RNP and donor DNA, why “The cuvette was then

incubated immediately at 16°C for 1 h”？Is this more efficient than conventional method （e.g., RT for 10 min）？

***This incubation step was adopted from the protocol of Shamoto et al. (Cells 2018, 7, 124; doi:10.3390). We have not tested the effect of varying this parameter, because the protocol (https://www.protocols.io/view/tim-a-targeted-insertional-mutagenesis-method-util-bp2l6n75rgqe/v1), which was optimized by Picariello, Hou et al. (PLoS ONE 2020, 15(5): e0232594), includes this step and already results in very efficient targeted insertional mutagenesis, which is the basis for the TIM-tagging method described here.

6. In line 114 “plasmid pLF5CsfGFP was first constructed by piecing together four fragments using NEBuilder® HiFi DNA Assembly Master Mix (catalog number E2621)”. The readers do not have idea how the author constructed the vectors. Please make it clearer that why these four fragments can be assembled when the authors designed the primers to amplify each fragment? It is better to make schematic representation as one of the supplementary figures.

***We thank the reviewer for this suggestion. A new supplementary figure (S1 Fig A and B) is now included to illustrate how plasmids pLF5CsfGFP and pLF5HA were constructed.

7. It is likely that Figure 1 is as same as the image appeared in PLoS One. 2020 May 13;15(5):e0232594. For the schematic representation data, such Figure 1-4 should move to supplementary figures.

***As pointed out in the legend to our original Figure 1, the figure was reproduced with modification from Figure 1 of PLoS One. 2020 May 13;15(5):e0232594; it was included in the Introduction to provide important background for the reader. However, this background is now provided by new Figure 1, which diagrammatically compares the key features of conventional insertional mutagenesis, TIM, and TIM tagging. As a result, old Figure 1 has been deleted. Figures 2-4 are novel figures specifically designed to illustrate the strategy of the TIM-tagging method and are essential for understanding the approach and the results in the subsequent Figures 5, 8, 9, and 10. Consequently we have kept them in the main text.

Reviewer #2: When submitting a paper to a journal, it is recommended that you standardize the format of the document according to the regulations of the journal, and edit the document within the level allowed by the journal so that reviewers can understand your research results more easily. Please next time should be more clear.

***We have reformatted the manuscript in the style recommended by the journal.

Reviewer #3: The authors performed endogenous gene editing of Chlamy. After RNP cleavage, the donor DNA (including homologous sequences, antibiotic resistance genes, tag genes, etc.) is introduced through homologous recombination, and the endogenous genes are tagged. In this paper, two genes were studied, one is LF5, which controls the length of the flagella, and tags HA at its C-terminus, and the other is NAP1L1, whose function is unknown, tags FLAG at its N-terminus.

All studies are based on the team's previous TIM (targeted insertional mutagenesis) design. The idea is: use RNP to cut the endogenous gene and introduce the donor DNA containing the antibiotic resistance gene (there is a homologous recombination arm on the left and right sides). The cutting pad is inserted through the homologous recombination arm.Two primers P1/P2 were designed at the endogenous gene cleavage site to identify the positive transformants. If the cleavage site is not inserted, the amplified fragment is a small fragment containing only part of the endogenous gene; if the cleavage site is inserted, the amplified fragment contains the endogenous gene and the resistance gene, and is a large fragment.

question:

1. In Figure 5, the authors performed sequencing analysis on 6 transformed strains (L1-L6) that amplified long fragments. The text shows that the sequence contains resistance genes, but the resistance genes are not marked in the S1 Appendix of the sequencing results. In theory, primers P1/P2 should amplify 563bp small fragments and large fragments in one reaction, but there is no small fragment and large fragment coexisting in one lane in the figure, how to explain?

***We thank the reviewer for pointing out that sequence encoding the antibiotic-resistance cassette was not marked in S1 Appendix; this sequence is now annotated for each PCR product shown in S1 Appendix. In addition, we have now annotated the antibiotic-resistance cassette in S2 and S4 Appendixes.

The reviewer is correct that one small fragment (563 bp) and one large fragment (3576 bp) would be predicted as products resulting from PCR amplification of the tagged gene in the experiments shown in Figure 5, yet only the small fragment was observed. There are several reasons for this. The primary reason is that we used an elongation time optimal for 563-bp products, so there would not have been enough time for the polymerase to amplify products as large as 3576 bp. Second, to facilitate screening of a large number of clones, we used crude DNA as template, and in our experience it is more difficult to amplify longer PCR products from crude DNA. Third, due to the high GC content of Chlamydomonas DNA, it is harder to amplify long PCR products than short ones. To clarify this for the reader, we have added the above information to the legend for Figure 5. We also added the sizes of PCR products L1-L6 to Figure 5 and to S1 Appendix.

To prevent confusion, we now point out (ll. 448 - 452) that the long PCR products L1-L6, the sizes of which range from 1235 bp to 1830 bp, are different from the large fragment (3576 bp) expected from an ideal integration of donor DNA. L1 to L6 were the products of alleles in which only a part of the donor DNA, consisting mainly of the drug-resistance cassette, had integrated into the gene. The annotations of the antibiotic-resistance cassette now added to S1 Appendix also help to clarify this. 

2. In Figure 9, 10 NAP1L1 transformed strains expressed FLAG-tagged proteins. Question: B12 is of S3 type (deletion mutation and point mutation), but its FLAG expression is very high, why? Why do some S1 and S2 not express tagged proteins? All have been sequenced, and they should all be expressed.

***These are intriguing questions for which we do not have definitive answers. The indel in strain B12 is at the 5’ end of the left homology arm of the inserted donor DNA. In the case of a simple integration as illustrated in Fig 10, this indel is well upstream of the start codon of the tagged gene and therefore would not be expected to affect its coding region. However, during donor DNA integration, in addition to precise integration events, other events such as large deletions, insertions, duplications, and conversions may happen in some strains. Here we sequenced only two small regions surrounding the ends of the integrated donor DNA. It is possible that a more complicated event happened between these two regions with a resulting effect on the regulation of expression. This speculation is now added to the Discussion, ll. 667 - 676. 

All of the strains marked S1 and S2 in Figure 8 (these clones are now marked SD and SE, respectively, in revised Figure 8) have precise integration at the 3’ end of the donor DNA. However, for those S1 and S2 strains that did not express the tagged proteins (strains B11, D3, and F11 in Figure 8), the integration of the 5’ end of the donor DNA apparently was accompanied by complicated events such as those listed above for strain B12, as evidenced by the absence of PCR products in Fig 9. In each case, the more complicated event presumably abolished expression of the tagged protein. The three strains are now labeled in Figure 9 and the above explanation is now added to the Results section (ll. 598 - 604).

3. S1-S10 marked in Figure 9 are easily confused with S1-S6 in Figure 8, so it is better to change the marking method.

***We thank the reviewer for pointing out that our original labeling in Figures 8 and 9 was confusing. In Figure 8, we have changed S1 to SD (indicating that the PAM region sequence is from the donor DNA), S2 to SE (indicating that the PAM region sequence is from the endogenous DNA), S3 to SX1, S4 to SX2, S5 to SX3, and S6 to SX4. These labels are now distinct from, and will not be confused with, those for Figure 9, which remains unchanged except for the addition of labels for strains B11, D3, and F11.

Overall, the study is relatively complete. However, since it is the establishment of the methodology, the analysis of the edited gene sequences is insufficient, and some sequences are not annotated. It is possible that the authors focused on the expression of tagged proteins, thus weakening the sequence changes after gene editing. Another important issue is that many of the resistant transformants in this study did not undergo gene editing as expected, and the resistance gene was inserted elsewhere. So, is there some way to determine that resistance genes etc are inserted only in the gap created by the RNP and not elsewhere. Otherwise, there may be unexpected variation in phenotype if mutations are made in this way.

***We thank the reviewer for asking for more sequence analysis and annotation. We previously sequenced and analyzed the region surrounding the integration site at the 5’ end of the donor DNA for the LF5-HA tagging project. We have now sequenced the region surrounding the integration site at the 3’ end of the donor DNA in the strains that express HA-tagged LF5; these data are now included as a new S2 Appendix. We also have now annotated the drug-resistance cassette sequences in S1, S2, and S4 Appendixes. The annotations will help interested readers follow the details of the integration events. 

With regard to the second issue, the reviewer is correct that insertions may occur in other places on the genome, and that this may cause unexpected variation in phenotype. This is a common concern for all transformation experiments using a selectable marker. There are ways – e.g., whole-genome sequencing or Southern blotting -- to determine if the resistance gene has inserted only in the gap created by the RNP and not elsewhere. However, in the case of TIM tagging, we were easily able to obtain multiple tagged strains, so it was more practical to study the different tagged strains to determine if they retained a wild-type phenotype and if the tagged protein behaved the same in the different strains. A strain that did not pass these checks would indeed be suspect, and therefore be discarded or crossed to a wild-type strain to determine if the unexpected phenotype segregated from the tagged gene. We now discuss these points in more detail than previously (ll. 677-686).

This manuscript is a further extension of the article published by their research group in Plos one in 2020. It is innovative and recommended to be published after revised.

Reviewer #4: The manuscript titled, ‘Direct in situ protein tagging in Chlamydomonas reinhardtii utilizing TIM, a method for CRISPR/Cas9-based targeted insertional mutagenesis’, by Hou et al has ued the green chlorophyte Chlamydomonas reinhardtii, a rather popular model system for flagellar studies to tag endogenous genes in situ using the modified TIM strategy. Undoubtedly, they have shown that this technique does circumvent the disadvantages of conventional approaches, the latter being more random. The authors have used this strategy on two genes: LF5/CDKL5, and Cre09.g416350/NAP1L1, both have not been studied previously in C. reinhardtii. The correct editing of these genes exhibit the success of the strategy (albeit at lower efficiencies).

I have only the following comments to make:

How equally useful is this strategy for other nuclear genes in C. reinhardtii?

***We chose two genes to tag to test the TIM-tagging approach, and it worked on both genes. We therefore expect that it will be equally useful for tagging most, if not all, other nuclear genes, including essential genes. The only factor that limits its usage is whether there is an appropriate RNP cut site near and upstream of the intended tagging site in the case of C-terminal tagging, or near and downstream of the intended tagging site in the case of N-terminal tagging. Appropriate cut sites for wild-type SpCas9, which was used in our study, are common in the C. reinhardtii genome. Moreover, if an appropriate wild-type SpCas9 cut site is not present, near-PAMless engineered CRISPR-Cas9 variants or different Cas endonucleases that recognize other PAMs are available, providing more options for the investigator. Thus, we do not anticipate significant limitations in applying TIM-tagging to other nuclear genes. We have now incorporated the above information in the Discussion and Conclusions (l. 783-790 and 814-817).

What about the sfGFP in the final construct (lines 122-124)? Is it cleaved off?

***sfGFP was not included in the PCR product amplified by LF5-20/LF5-31, and thus was not in the final construct (pLF5HA). To clarify this for the reader, we have added diagrams (S1 Fig A and B) to illustrate how the final construct was created.

Although the writing is simple and lucid, the authors have used three strains and must make mention of the type of strain they use in the M/M or wherever possible. It can get confusing, at times.

***Thank you for pointing out that our listing of the strains, and identification of their uses, was confusing. In the first paragraph of the Materials and Methods, we now describe the strains we used and what they were used for. We have added two strains used for preparation of autolysin. In the course of revising the manuscript, we found that we had erroneously listed one strain not used; mention of that strain has been deleted. Finally, we have now identified, in the legends for Figs 5, 6, 7, 11, and S1, the wild-type strain (g1) referred to there. We trust that these changes will preclude any confusion. 

The authors have defended their modified TIM-based strategy to be relatively simple and quick, as opposed to the conventional approaches. A comparison in terms of time and ‘simplicity’ (whatever that implies) is required – especially since they claim this to be done for the first time in Chlamydomonas.

***By saying “relatively simple” (Introduction) and “carried out more quickly” (Discussion), we were referring to the fact that with TIM-tagging there is no need to create or obtain a null mutant to ensure that the tagged protein is expressed in the absence of untagged protein. To avoid any misunderstanding, we have deleted “the experimental procedure is relatively simple” from the Introduction and modified the sentence in the Discussion.

[Review comment]

PONE-D-22-20272

Decision : Minor Revision

This paper introduces the most advanced technology that can be used for functional gene research using genome editing in microalgae. The tagging method based on the homologous recombination along with the TIM has a great advantage in that it can tag while maintaining the sequence of the endogenous gene and its expression. Unfortunately, these innovations do not provide new insights because they have already been applied to other species. However, if it is considered restricted to microalgae, it can be evaluated that it could contribute to technological progress.

It is required that more detailed information be supplemented in order to share more accurate technological innovation with readers through this paper.

Minor Review

1. Figure 1 should be re-draw in order to TIM and TIM-tagging methods can be understood conceptually. It should be different from ref [20].

***We thank the reviewer for this suggestion. To provide readers with a conceptual understanding of the TIM method and how TIM-tagging is built upon that advance, we now provide a new Figure 1 that diagrammatically compares the key features of conventional insertional mutagenesis, TIM, and TIM tagging. We have revised relevant parts of the Introduction accordingly.

2. Since this research is intended as an introduction to a novel gene editing strategy, a more detailed explanation is needed to help readers understand the TIM-tagging method. Add the structural description of TIM-tagging in the introduction or discussion.

***We agree that a more detailed explanation of TIM-tagging is justified to help readers understand this novel gene editing strategy. New Figure 1C with accompanying text now provides a structural description of the method’s key genetic features and their relationship to one another.

3. Add the fluorescence image of mNeongreen in fig 11 or thereafter.

***As described above in answer to Reviewer #1’s comment 4, we have been unable to detect mNeonGreen signal in living cells or FLAG signal in fixed cells, and so could not determine the location of NAP1L1 within the cell. Although the tagged protein can be detected by western blotting, it apparently is not expressed at a level sufficiently high to be detected at the single-cell level with our microscopes. This is now clarified in the Results section (ll. 613-616).

---

## [Decision Letter · Decision Letter 1]

28 Nov 2022

Direct in situ protein tagging in Chlamydomonas reinhardtii utilizing TIM, a method for CRISPR/Cas9-based targeted insertional mutagenesis

PONE-D-22-20272R1

Dear Dr. Witman,

We’re pleased to inform you that your manuscript has been judged scientifically suitable for publication and will be formally accepted for publication once it meets all outstanding technical requirements.

Kind regards,

Hodaka Fujii, M.D., Ph.D.

Academic Editor

PLOS ONE

Additional Editor Comments (optional):

Reviewers' comments:

Reviewer's Responses to Questions

**Comments to the Author**

1. If the authors have adequately addressed your comments raised in a previous round of review and you feel that this manuscript is now acceptable for publication, you may indicate that here to bypass the “Comments to the Author” section, enter your conflict of interest statement in the “Confidential to Editor” section, and submit your "Accept" recommendation.

Reviewer #1: All comments have been addressed

Reviewer #3: All comments have been addressed

Reviewer #4: All comments have been addressed

2. Is the manuscript technically sound, and do the data support the conclusions?

Reviewer #1: Yes

Reviewer #3: Yes

Reviewer #4: Yes

3. Has the statistical analysis been performed appropriately and rigorously? 

Reviewer #1: Yes

Reviewer #3: Yes

Reviewer #4: Yes

4. Have the authors made all data underlying the findings in their manuscript fully available?

Reviewer #1: Yes

Reviewer #3: Yes

Reviewer #4: Yes

5. Is the manuscript presented in an intelligible fashion and written in standard English?

Reviewer #1: Yes

Reviewer #3: Yes

Reviewer #4: Yes

6. Review Comments to the Author

Reviewer #1: (No Response)

Reviewer #3: Dear editor.

In this round of manuscript review, the author explained and reasonably answered the questions raised previously, and added some questions explanations to the discussion part of the manuscript. In addition, the author revised some figures and supplemented some data. This is worthy of recognition and appreciation. So I agree to accept the publication.

Reviewer #4: The revised version of the manuscript titled, 'Direct in situ protein tagging in Chlamydomonas reinhardtii utilizing TIM, a method for CRISPR/Cas9-based targeted insertional mutagenesis' by Hou et al has improved considerably and has addressed all the queries addressed by me.

I have no further queries and the manuscript may be accepted as it is now easier to understand and suitable for publishing in PlosOne.

7. PLOS authors have the option to publish the peer review history of their article (what does this mean?). If published, this will include your full peer review and any attached files.

Reviewer #1: No

Reviewer #3: No

Reviewer #4: No

---

## [Editor Report · Acceptance letter]

1 Dec 2022

PONE-D-22-20272R1 

Direct in situ protein tagging in *Chlamydomonas reinhardtii* utilizing TIM, a method for CRISPR/Cas9-based targeted insertional mutagenesis 

Dear Dr. Witman:

I'm pleased to inform you that your manuscript has been deemed suitable for publication in PLOS ONE. Congratulations! Your manuscript is now with our production department. 

Kind regards, 

on behalf of

Dr. Hodaka Fujii 

Academic Editor

PLOS ONE